# Soluble guanylate cyclase signalling mediates etoposide resistance in progressing small cell lung cancer

Maximilian W. Schenk[1], Sam Humphrey[1], A. S. Md Mukarram Hossain[1], Mitchell Revill[1], Sarah Pearsall[1], Alice Lallo[1], Stewart Brown[1], Samuel Bratt[1], Melanie Galvin[1], Tine Descamps[1], Cong Zhou[1], Simon P. Pearce [1], Lynsey Priest[1], Michelle Greenhalgh[2], Anshuman Chaturvedi[2], Alastair Kerr [1,3], Fiona Blackhall[2,3,4], Caroline Dive [1,3,5✉] & Kristopher K. Frese [1,3,5]

Small cell lung cancer (SCLC) has a 5-year survival rate of <7%. Rapid emergence of acquired resistance to standard platinum-etoposide chemotherapy is common and improved therapies are required for this recalcitrant tumour. We exploit six paired pre-treatment and post-chemotherapy circulating tumour cell patient-derived explant (CDX) models from donors with extensive stage SCLC to investigate changes at disease progression after chemotherapy. Soluble guanylate cyclase (sGC) is recurrently upregulated in post-chemotherapy progression CDX models, which correlates with acquired chemoresistance. Expression and activation of sGC is regulated by Notch and nitric oxide (NO) signalling with downstream activation of protein kinase G. Genetic targeting of sGC or pharmacological inhibition of NO synthase re-sensitizes a chemoresistant CDX progression model in vivo, revealing this pathway as a mediator of chemoresistance and potential vulnerability of relapsed SCLC.

---

[1] Cancer Research UK Manchester Institute Cancer Biomarker Centre, University of Manchester, Alderley Park, Macclesfield SK10 4TG, UK. [2] Christie National Health Service Foundation Trust, Division of Cancer Sciences, The University of Manchester, Manchester M20 4BX, UK. [3] Cancer Research UK Lung Cancer Centre of Excellence at the University of Manchester, Oxford Road, Manchester M13 9PL, UK. [4] Division of Cancer Sciences, Faculty of Biology, Medicine and Health, University of Manchester, Manchester M13 9PL, UK. [5] These authors jointly supervised this work: Caroline Dive, Kristopher K. Frese. ✉email: caroline.dive@manchester.ac.uk

Small cell lung cancer (SCLC) is an aggressive neuroendo-crine tumour accounting for ~250,000 deaths each year[1]. Poor prognosis is attributed to diagnosis of most patients with metastatic disease and typically rapid disease relapse within <1 year of diagnosis, usually after an initial response to platinum-etoposide doublet chemotherapy alone[2] or, more recently, in combination with immune checkpoint inhibitors[3]. The majority of clinical trials of targeted therapies for SCLC patients have failed, where this heterogeneous disease is treated uniformly in the absence of predictive biomarkers.

A major barrier to discovery of effective therapies that can be implemented after first-line chemotherapy fails is an insufficient understanding of what drives the rapid acquisition of chemoresistance despite initial chemosensitivity. The paucity of tumour biopsies at disease progression and consequently of patient-derived preclinical models that faithfully represent the biology of the patient's progressing tumour has hampered research with some notable exceptions. Profiling of 12 paired patients' tumour samples at diagnosis and relapse identified acquired mutations in Wnt-pathway genes[4] and patient-derived xenograft (PDX) studies where chemoresistance was generated by repeated treatment of mice with chemotherapy led to the discovery of schlafen family member 11 (SLFN11) and enhancer of zeste homology 2 (EZH2) as acquired chemoresistance mediators[5]. Research on genetically engineered mouse models (GEMMs) and PDX reported increased MYC expression during disease progression[6,7].

A recent consensus report defined four subtypes of SCLC based on transcription factor (TF) expression: two neuroendocrine (NE) subtypes expressing TFs achaete-scute complex homolog-like 1 (ASCL1) and neurogenic differentiation factor 1 (NEUROD1), and two non-neuroendocrine (Non-NE) subtypes expressing yes-associated protein 1 (YAP1) and POU class 2 homeobox 3 (POU2F3)[8,9]. We subsequently reported an additional transcriptionally distinct NE subtype based on the expression of atonal BHLH transcription factor 1 (ATOH1)[10]. The importance of inter- and intra-tumoural heterogeneity (ITH) as a mechanism of acquired chemoresistance in SCLC[11] has been suggested, but the precise relationships between the subtypes and chemoresistance remain unclear, though several studies demonstrate that in vitro, Non-NE cells are more chemoresistant than NE cells[9,12].

In 2014, we described a different approach to generate patient-faithful preclinical mouse models exploiting the relatively prevalent circulating tumour cells (CTCs) in patients with SCLC[13]. These models, which we termed CDX, mirror the donor patients' response to chemotherapy and offer the opportunity to collect blood samples before and after drug-resistant relapse to generate longitudinal models of clinically acquired drug resistance[14]. We subsequently described short-term cultures from disaggregated CDX tumours that could be genetically modified, enabling experiments to validate candidate genes responsible for acquired drug resistance[15]. In the current study, we examined models derived before an SCLC patient's treatment and again after their treatment at disease progression from six individual patients[10]. We found that both subunits of a soluble guanylate cyclase (sGC), a protein important for sensing nitric oxide (NO), were recurrently upregulated in disease progression vs. baseline CDX models. This upregulation correlated with acquired chemoresistance of CDX in vivo and genetic reduction of sGC levels or pharmacological inhibition of sGC activity with a nitric oxide synthase (NOS) inhibitor re-sensitized a progression CDX model to cisplatin/etoposide in vivo. These data implicate the sGC signalling pathway as a potentially targetable vulnerability of relapsing SCLC.

## Results

### Genomic characterization of clinical resistance models. To enhance understanding of SCLC acquired drug resistance, one

major ambition of the NCI Recalcitrant Cancer Act (2012) was generation of models derived from paired biopsies obtained from patients with SCLC prior to therapy and again after disease progression[16]. The clear advantage of the CDX approach to longitudinal models is that only a routine 10 ml blood draw rather than an invasive biopsy is required. We derived paired pre-treatment and post-progression CDX models from six patients and performed whole-exome sequencing (WES) and RNA-sequencing (RNA-seq) (Fig. 1a and ref. [10]). All six patients had an initial partial or stable response to their chemotherapy (Supplementary Table 1) followed by disease progression <1 year later, typifying the aggressive course of this disease[10]. Our paired CDX models represented ASCL1 (CDX3/3P, CDX18/18P, CDX20/20P, CDX42/42P), NEUROD1 (CDX8/8P) and ATOH1 (CDX17/17P) subtypes of SCLC[10]. Previous publications studying acquired chemoresistance in SCLC have described both genomic alterations[4] and changes in the transcriptome[5–7,11] as drivers for acquired chemoresistance. Thus, our initial attention focused on genomic changes acquired through SCLC progression by performing WES of the paired models. Our analysis identified typical SCLC-associated mutations[17] with TP53 and RB1 the most frequently mutated genes in our baseline and progression models (Fig. 1b). Of note, while we did not detect TP53 and RB1 mutations in CDX models 18 and 20 using a pipeline reporting only somatic mutations, we did subsequently identify germline mutations in their donors (missense mutations for TP53 in CDX18/18P and CDX20/20P donors and nonsense RB1 mutations for the CDX20/CDX20P donor). Similar to Gardner et al.[5], mutations were conserved through acquisition of chemoresistance in the majority of cases across the paired CDX models. To explore the likely identity of passenger vs. driver mutations, we sought recurrently acquired mutations shared between the progression models (Fig. 1c). We detected only three mutations that were shared between multiple progression models: F-box protein 10 (FBXO10), cilia and flagella associated protein 47 (CFAP47) and dysferlin (DYSF), all mutations predicted to be passenger mutations[18]. We also assessed the number of shared and private mutations between each individual patient's baseline and progression models to explore further their potential as mechanisms of acquired chemoresistance (Supplementary Fig. 1a).

The majority of mutations were shared between a patient's baseline and progression models; however, there was a trend ($p = 0.06$) of increased private mutations in progression models implicating acquisition of mutations through either SCLC progression or a mutagenic effect associated with exposure to chemotherapy. Next, we examined whether mutational signatures changed between individual baseline and progression models (Supplementary Fig. 1b, c). As expected, the most frequent mutational signature was signature 4, associated with C>A mutations and tobacco smoking reflecting the patients' smoking histories. One exception was CDX models derived from patient 20, which displayed low signature 4 mutations. Clinical records revealed that this patient received a sibling allogenic stem cell transplant for the treatment of acute myeloid leukaemia (AML), which could explain the low smoking related signature. Mutational signatures were broadly conserved from baseline to progression. We also addressed whether there were changes in chromosomal instability during disease progression by analysing copy number (CN) changes between baseline and progression models (Fig. 1d). Consistent with our previous study of longitudinal sampling and analysis of CTCs[19], we did not identify recurrent global changes between CN profiles of progression and baseline models. In conclusion, these results are concordant with those of others[5–7,11], indicating that genomic alterations alone are unlikely to account fully for emergence of acquired resistance in our progression models. We refocused our

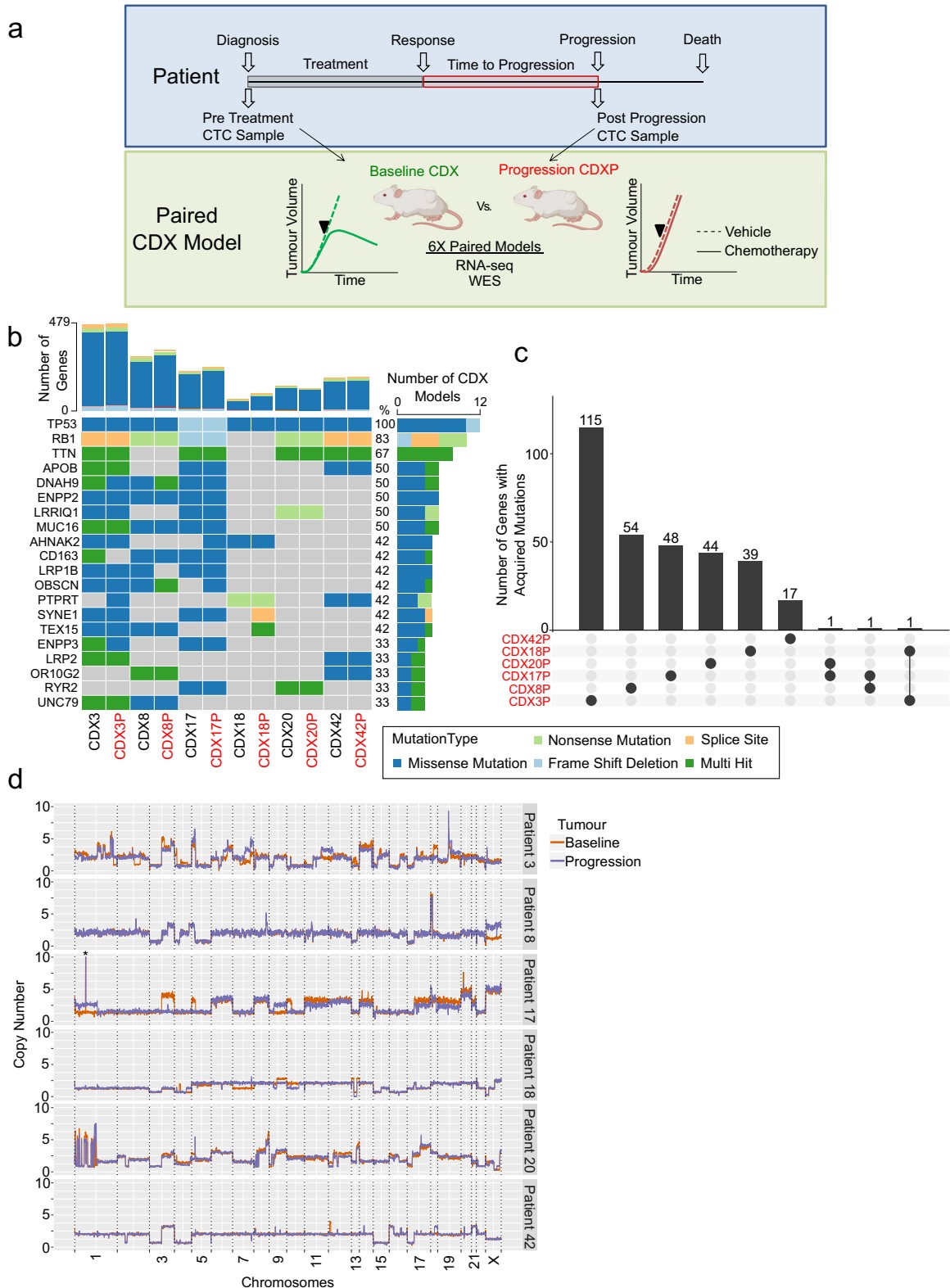

studies to investigate changes in the transcriptomes of pre- and post-chemotherapy CDX models.

**sGCs are upregulated during disease progression and correlate with acquired rather than inherent chemoresistance in SCLC.** Previous studies suggested that acquired chemoresistance in SCLC can be driven through acquisition of transcriptional changes rather than emergence of recurrent mutations[5,19].

Principal component analysis and analysis of inter-sample similarity of RNA-seq data from the six paired CDX models revealed clustering of individual pairs, confirming their origin from the same patient and that the transcriptomes of progression models more closely resemble their baseline counterpart rather than other progression models (Supplementary Fig. 2a, b). First, we examined previously reported mechanisms of acquired chemoresistance. Transcriptional downregulation of the putative

**Fig. 1 Genomic characterization of clinical resistance models. a** Strategy to generate paired models of acquired chemoresistance. CTCs were isolated from SCLC patients pre-treatment and at post-chemotherapy disease progression and CDX models were generated. **b** The most frequently mutated genes are shown for the CDX baseline and progression pairs from WES. Presence of a particular mutation is colour coded by the mutation type. The top panel represents the number of genes where at least one mutation was found, and the right panel shows the number (and percentage) of models with mutations detected for a gene. **c** Number of genes with acquired mutations in CDX progression models. Bars on the top panel show the number of genes with acquired mutations. Lines with solid circles on the bottom panel below each bar represent the CDX models that share the acquired mutations. The shared acquired mutations are F-box protein 10 (*FBXO10*), cilia, and flagella-associated protein 47 (*CFAP47*) and dysferlin (*DYSF*). **d** Copy number (CN) changes in chromosomes of baseline and progression CDX models. Copy numbers were estimated from the baseline and progression tumour samples with respect to their germline samples from WES data. *Value in chromosome 1 of patient 17 goes beyond the *y*-axis limit. See also Supplementary Fig. 1.

RNA-DNA helicase *SLFN11* (ref. [5]) was observed in CDX17P ($p = 0.045$); however, this was not observed in any other CDX pair (Supplementary Fig. 2c). The previously reported upregulation of Wnt-pathway genes at disease progression[4] was not observed in any of the progression CDX (Supplementary Fig. 2d).

Next, we examined genes recurrently upregulated in progression compared to pre-treatment CDX. Gene set enrichment analysis of pathways recurrently up- and downregulated in our progression models identified cell cycle and ribosome biogenesis pathways as some of the most upregulated pathways, essential for sustaining tumour growth and proliferation (Supplementary Fig. 2d). Both subunits of an sGC, guanylate cyclase soluble subunit alpha-1 (*GUCY1A1*) and beta-1 (*GUCY1B1)*, essential components of the nitric oxide (NO) signalling pathway, were the most significantly recurrently upregulated genes in our post-chemotherapy disease progression models ($p < 0.001$, Fig. 2a, Supplementary Data 1). This was not due to amplification of *GUCY1A1* or *GUCY1B1* according to our matched WES data (Supplementary Fig. 2g). sGCs are the primary molecular sensors of NO with a key role in NO signalling to regulate several physiological processes[20]. Of note, co-operation of phosphatidylinositol 3-kinase (PI3K)/AKT, Notch, and NO signalling has been reported during tumourigenesis in a *Drosophila* eye cancer model[21] and NO can have both pro- and anti-tumourigenic effects, depending on NO levels and cancer types[22,23]. Furthermore, Notch signalling contributes to an NE to Non-NE phenotypic switch, the latter displaying relatively decreased chemosensitivity[12,24]. These results prompted our investigation of the role of NO signalling and sGCs in SCLC progression. We focused our initial studies on GUCY1B1 as opposed to GUCY1A1, since the former contains the NO-sensing H-NOX domain essential for activation of the protein[25]. First, we confirmed the upregulation of GUCY1B1 protein expression in CDX8P, CDX17P, CDX18P and CDX20P using immunohistochemistry (IHC) and western blot (Fig. 2b, c and Supplementary Fig. 2e). Next, we asked whether an increase in GUCY1B1 protein expression correlated with acquired chemoresistance in our CDX models. Comparison of CDX in vivo chemoresponse to GUCY1B1 expression revealed a trend for progression CDX models with upregulated GUCY1B1 expression responding less well to chemotherapy (Fig. 2d, $r^2 = 0.462$, $p = 0.137$).

When we assessed *GUCY1B1* gene expression across a wider panel of baseline CDX models (derived from patients before their treatment), there was no correlation between *GUCY1B1* expression and the subsequent chemotherapy response of patient donors, suggesting that pre-treatment sGC expression levels do not signpost inherent, chemorefractory disease (Fig. 2e).

To study the relationship between GUCY1B1 and inherent chemoresistance further and to determine whether GUCY1B1 expression is associated with disease stage and clinical outcomes of SCLC patients, we examined GUCY1B1 expression by IHC across a tumour microarray (TMA) generated from 54 treatment-naive patients (Supplementary Fig. 2f). In evaluable patients, GUCY1B1 expression was not associated with disease stage ($p = 0.124$), although there were relatively few tumours from extensive stage patients represented on the TMA (7 vs. 37, Fig. 2f) reflecting the challenge of obtaining tumour tissue from extensive stage patients. There was no significant difference in GUCY1B1 expression between the 26 patients who initially responded to chemotherapy and the eight patients who did not ($p = 0.571$) (Fig. 2g). Furthermore, GUCY1B1 expression in pre-treatment SCLC was not associated with survival (HR [95% CI] = 1.02 [0.98, 1.05], $p = 0.438$) in this exploratory dataset of the 33 cases with available outcome data, using IHC scores dichotomized at a score of 12 as the cut-point (Fig. 2h and see 'Methods'). Our results suggest that sGCs are upregulated during SCLC progression and that GUCY1B1 upregulation is a mechanism of acquired, as opposed to inherent, chemoresistance.

**Regulation of sGC subunit expression by Notch signalling in SCLC.** The expression of both sGC subunits is regulated by Notch signalling in human endocardial cells[26] and NO synthase (NOS) fuels tumourigenesis by activated Notch in a *Drosophila* eye cancer model[21]. Upon binding of NO, sGC catalyses the conversion of guanosine triphosphate (GTP) to the second messenger cyclic guanosine monophosphate (cGMP), where it mediates a variety of downstream effects (Fig. 3a and ref. [20]). We previously reported the upregulation of Notch receptors in four out of six progression pairs[10]. Furthermore, two negative regulators of Notch signalling, BEN domain containing 6 (*BEND6*)[27] and delta-like 1 homolog (*DLK1*)[28] were downregulated in the CDX progression models (twofold and threefold, respectively, $p < 0.001$, Supplementary Data 1) consistent with elevated Notch pathway activity, so we asked whether Notch could regulate sGC subunit expression in SCLC. A panel of human SCLC cell lines was assessed for GUCY1B1 expression levels to select those appropriate for pathway analysis and function testing studies (Fig. 3b). To determine whether GUCY-expression was Notch-dependent, we treated Non-NE H196 cells with the γ-secretase inhibitors DAPT and DBZ. DAPT and DBZ treatment inhibited Notch activity, measured by a decrease in the Notch target gene hes related family bHLH transcription factor with YRPW motif 1 (*HEY1*) ($p < 0.001$) and concomitantly led to reduction in sGC subunit transcripts (*GUCY1B1* $p < 0.001$, *GUCY1A1* $p = 0.009$) (Fig. 3c and Supplementary Fig. 3a). DAPT and DBZ also reduced protein expression of both the Notch target hes family bHLH transcription factor 1 (HES1) ($p = 0.099$) and GUCY1B1 ($p = 0.078$) (Fig. 3d and Supplementary Fig. 3b). In order to confirm the impact of pharmacological inhibition of Notch signalling on downstream effectors of the pathway, we downregulated Notch1 genetically using CRISPR. Downregulation of Notch1 expression reduced expression of the Notch target gene HES1 as well as the expression of GUCY1B1 consistent with the effects of DAPT and DBZ (Fig. 3e).

To determine whether Notch activation is sufficient to induce sGC expression, the Notch1 intracellular domain (N1ICD) was overexpressed in Non-NE H1048 cells which have low basal levels of GUCY1B1. Overexpression of N1ICD significantly induced Notch

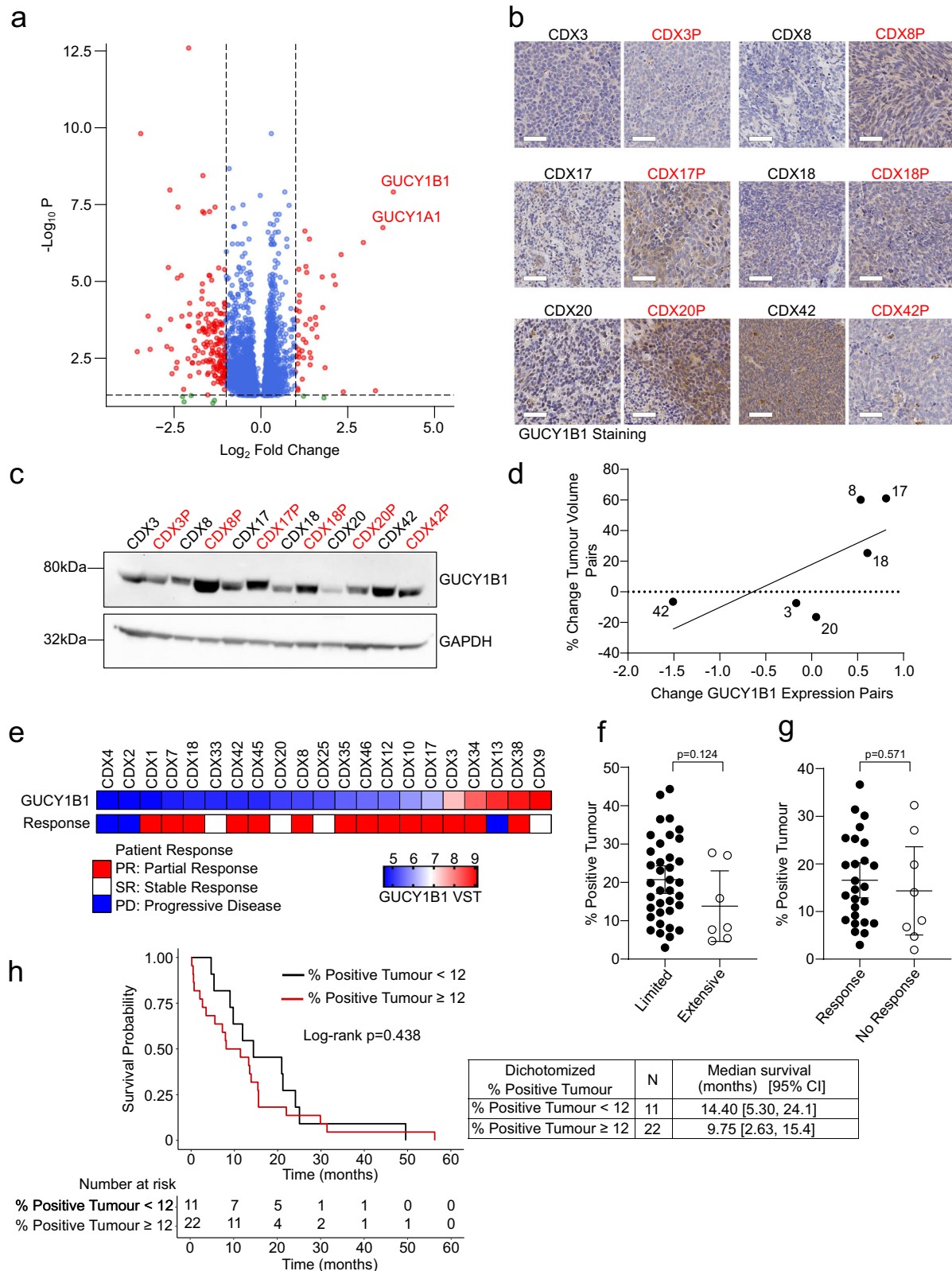

signalling, shown by an increase in HES1 ($p < 0.001$) and GUCY1B1 ($p = 0.005$) expression (Fig. 3f). We sought to confirm and extend these results obtained from established cell lines in CDX ex vivo cultures, generated by disaggregation of CDX tumours, and short-term culture of derived human cells[15]. Endogenous Notch pathway signalling promotes an NE to Non-NE fate switch in a SCLC GEMM[12]. CDX17 and CDX17P ex vivo cultures of NE and Non-

NE cell subpopulations were generated by separating floating cell aggregates and adherent cells, respectively, and subpopulations confirmed by the expression of the NE marker SYP[9,29]. As expected, GUCY1B1 expression was elevated in CDX17P compared to CDX17 and GUCY1B1 expression was higher in the Non-NE cell subpopulation in both models (Fig. 3g). Treatment of CDX17P Non-NE cells with DAPT or DBZ reduced *GUCY1A1* ($p = 0.002$)

**Fig. 2 sGCs are upregulated during disease progression and correlate with acquired rather than inherent chemoresistance in SCLC. a** RNA-seq of six CDX progression models showing recurrent differentially expressed genes. Volcano plot indicates recurrent significantly ($padj < 0.05$) upregulated ($log2FC > 1$) and downregulated ($log2FC < -1$) genes in six progression models vs. corresponding pre-treatment models. Genes that passed only one of these thresholds are indicated in blue/green, $p$ values from negative binomial distribution, calculated by DESeq2. **b** IHC of paired CDX tumours for GUCY1B1 expression (brown stain). Scale bar set to 50 μm and equivalent throughout panels. Whole tumours were analysed and representative areas are shown. Representative images from one mouse per group are shown. **c** Western blot for GUCY1B1 expression in paired CDX progression tumour lysates. Blotting was performed on three animals per model ($n = 3$, Supplementary Fig. 2e) and a representative western blot is shown. **d** Correlation of change in GUCY1B1 protein expression and change tumour volume in progression and baseline CDX models after treatment with cisplatin/etoposide. $p = 0.137$, Pearson $R^2$ value $= 0.462$, from linear regression. **e** Comparison of donor patients' chemotherapy responses with GUCY1B1 expression (variance-stabilizing transformation, VST) of the corresponding CDX model determined by RNA-seq. **f** Comparison of GUCY1B1 TMA % positive tumour and stage of disease of treatment-naive patients. $n = 37$ limited and $n = 7$ extensive stage patients, data are represented as mean ± 95% confidence interval (CI). $P$ values from two-sided unpaired Student's $t$-test, $t = 1.570$, d.f. $= 42$. **g** Comparison of GUCY1B1 % positive tumour and response of treatment-naive patients. Response summarizes partial and complete response; no response stable and progressive disease. $n = 26$ response group and $n = 8$ no response group, data are represented as mean ± 95% CI. $P$ values from two-sided unpaired Student's $t$-test, $t = 0.573$, d.f. $= 32$. **h** Survival analysis of treatment-naive SCLC patients using dichotomized % positive tumour. Two-sided Kaplan–Meier (log rank) analysis was performed to estimate the median survival in each group. Wald test $= 0.85$, log-rank chi square value $= 0.62$, d.f. $= 1$, and $p$ value $= 0.438$. See also Supplementary Fig. 2.

and to a lesser extent *GUCY1B1* ($p = 0.055$) expression consistent with reduced Notch signalling (Fig. 3h and Supplementary Fig. 3c). Overexpression of N1ICD in CDX17P NE cells led to upregulation of *GUCY1A1* ($p = 0.005$) without a change in *GUCY1B1* ($p = 0.370$) (Fig. 3i). Furthermore, by performing a co-immunofluorescence assay on NE and Non-NE cells from CDX17 and CDX17P, we confirmed upregulation of Notch signalling in Non-NE cells (HES1 expression) concomitant with an upregulation of GUCY1B1 expression (Supplementary Fig. 3d, e). In order to address whether the sGC subunits are Notch target genes, we investigated binding of Notch1 to previously reported RBPJ-binding sites in the *GUCY1A1* and *GUCY1B1* promoter[26] in H1048 cells overexpressing N1ICD by performing chromatin immunoprecipitation (ChIP) followed by qPCR (Fig. 3j). As a positive control, we demonstrated N1ICD binding to a region in the *HES1* promoter ($p = 0.001$). Our data indicate binding of N1ICD to promoter regions of *GUCY1A1* ($p = 0.034$) and *GUCY1B1* ($p = 0.048$) substantiating both sGC subunits as direct Notch target genes.

Taken together, these studies indicate that sGC subunit expression in SCLC is regulated by Notch signalling.

**sGC signalling requires nitric oxide.** To study the impact of sGC signalling on SCLC cell fate, we reduced *GUCY1B1* expression using two different sgRNAs targeting *GUCY1B1*. Cells were treated with the NO donor DETA NONOate and the impact was assessed by phosphorylation of vasodilator-stimulated phosphoprotein (pVASP) levels, an established biomarker of sGC pathway activation[30]. DETA NONOate treatment increased VASP phosphorylation in non-targeting (sgNTA) control cell lines, whereas this effect was attenuated or completely abrogated in sg*GUCY1B1*-1 (sgB1-1) and sg*GUCY1B1*-2 (sgB1-2) cells, respectively (Fig. 4a). To verify the specificity of this effect, we demonstrated that the sGC-specific activator BAY 41-2272 phenocopies the effects elicited by DETA NONOate (Supplementary Fig. 4a). To validate these findings, we treated wild-type (WT) cell lines with 1H-[1,2,4]oxadiazolo[4,3-a]quinoxalin-1-one (ODQ), a NO competitive inhibitor of sGC activity. As expected, treatment with DETA NONOate increased sGC signalling, whereas combination of DETA NONOate with ODQ reduced sGC pathway activation to untreated levels (Fig. 4b).

**Impact of sGC signalling on SCLC cell fate and chemotherapy responses in vitro.** We focused initial efforts on studying the impact of sGC signalling on proliferation and migration/invasion, phenotypes previously reported as influenced via this pathway[31,32]. No difference in cell numbers was observed between sgB1-1, sgB1-2 and sgNTA cells (Supplementary Fig. 4b). Similarly, sgB1-1 and sgB1-2 cell lines did not exhibit

any difference compared to controls in their ability to migrate or invade towards a serum gradient (Supplementary Fig. 4c, d). Next, control sgNTA, sgB1-1 and sgB1-2 cell lines were tested for their response to cisplatin or etoposide after prior stimulation with DETA NONOate. sgB1-1 and sgB1-2 cells showed negligible change in cisplatin sensitivity but both cell lines were more sensitive to etoposide, with sgB1-2 cells showing the greater (2.5-fold increase) etoposide sensitivity than sgB1-1 ($p < 0.001$) (Fig. 4c). These data comparing sgB1-1 and sgB1-2 cells are consistent with pathway activation readouts; in sgB1-2 cells sGC activation by DETA NONOate was not detected, whereas sgB1-1 cells still showed partial pathway activation (Fig. 4a). As expected, there was no difference in chemosensitivity between sgNTA, sgB1-1 and sgB1-2 cell lines without prior stimulation with DETA NONOate (Supplementary Fig. 4e, f). Subsequently, we exposed WT cell lines to DETA NONOate alone or in combination with ODQ and treated cells with etoposide. Activation of sGC signalling with NO made cells almost eightfold more resistant to etoposide compared to ODQ treated cells ($p = 0.002$) (Fig. 4d, e). Combined treatment of cells with the sGC inhibitor rescued this effect, suggesting that elevated sGC signalling causes increased resistance to etoposide in H196 cells.

Previous studies have shown that NO can directly modify and inactivate etoposide[33,34]. To evaluate this specific mechanism of resistance, we repeated the experiment with doxorubicin, a drug with a different chemical structure that also inhibits topoisomerase II[35]. Cells became ~threefold more resistant to doxorubicin after treatment with NO, an effect that was rescued by treatment with ODQ (Fig. 4f, g), suggesting that etoposide resistance is mediated through increased sGC signalling rather than direct effects of NO on the etoposide molecule.

**sGC signalling via PKG reduces response to etoposide in vitro.** PKG is one of the major downstream effectors of sGC pathway activation[36]. PKG activation in vitro elicits chemoresistance in ovarian cancer[22] and in non-small cell lung cancer (NSCLC)[37]. Therefore, we hypothesized that sGC pathway activation could mediate drug resistance via PKG. To test this hypothesis, we treated WT cell lines either with DETA NONOate alone or in combination with a selective PKG inhibitor. As expected, treatment with DETA NONOate alone increased sGC signalling and PKG-dependent VASP phosphorylation, whereas combination with the PKG inhibitor reduced this effect (Fig. 4h). Concordantly, treatment with the PKG inhibitor reversed NO-dependent etoposide and doxorubicin resistance (Fig. 4i–l), suggesting that sGC pathway activation increases etoposide and doxorubicin resistance via PKG activation.

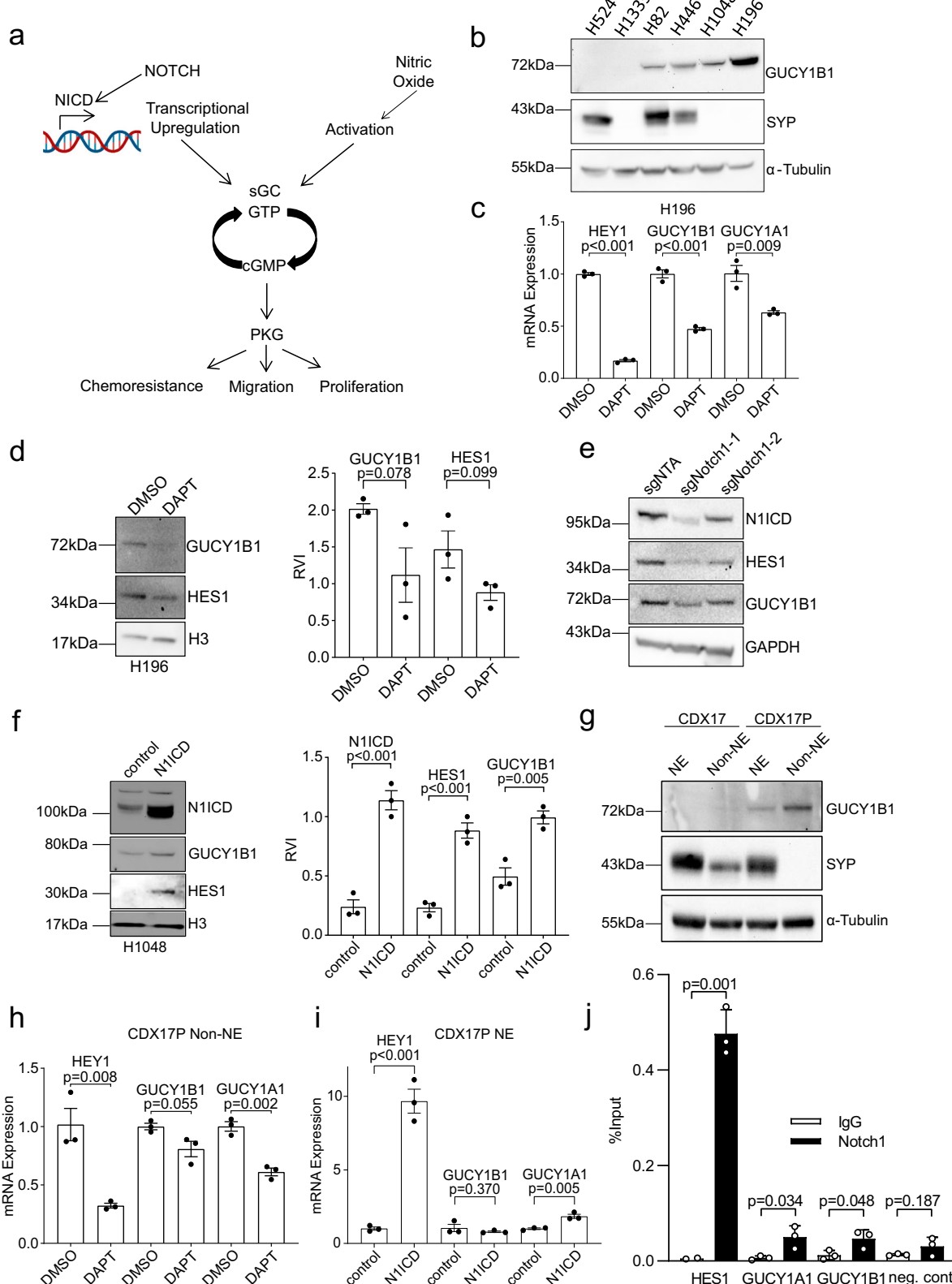

**Loss of sGC signalling sensitizes CDX17P to cisplatin/etoposide in vivo**. We sought to determine whether our in vitro findings would be recapitulated in vivo in CDX17P, a chemoresistant progression model with elevated GUCY1B1 expression. In the absence of a potent bioavailable sGC inhibitor for in vivo use, we adopted a CRISPR approach to target *GUCY1B1* in CDX17P ex vivo and then tested sensitivity to the SCLC standard of care chemotherapy

regimen of combined etoposide and cisplatin in vivo (Fig. 5a). Prior to the efficacy study, we implanted mice with CDX17P sgNTA and sgB1-2 cells to confirm that the decreased expression was maintained over the time course of the efficacy experiment. GUCY1B1 expression was compared in sgB1-2 and sgNTA tumours by IHC when they reached 700–800 mm³ showing that *GUCY1B1* loss was maintained in vivo for up to 41 days (also validating the specificity

**Fig. 3 Regulation of sGC subunit expression by Notch signalling in SCLC. a** Schematic of the working hypothesis. **b** Western blot for GUCY1B1 and synaptophysin (SYP) expression in SCLC cell lines ($n = 3$). **c** HEY1, GUCY1B1 and GUCY1A1 mRNA expression (RT-qPCR) of DMSO- or DAPT-treated H196 cells. $n = 3$, data are represented as mean ± SEM. $P$ values from two-sided unpaired Student's $t$-test. HEY1 $p = 0.0001$, GUCY1B1 $p = 0.0002$. **d** Western blot for GUCY1B1 and HES1 of DMSO- or DAPT-treated H196 cells. Quantification of GUCY1B1 (relative volume intensity, RVI) on the right, normalized to histone H3. $n = 3$, data are represented as mean ± SEM; $p$ values from two-sided unpaired Student's $t$-test. **e** Western blot for Notch1, HES1 and GUCY1B1 in sgNTA, sgNotch1-1 and sgNotch1-2 H1048 cells ($n = 3$). **f** Western blot for N1ICD, HES1 and GUCY1B1 in empty vector control and N1ICD-overexpressing H1048 cells. Quantification of GUCY1B1 (RVI) on the right, normalized to H3. $n = 3$, data are represented as mean ± SEM. $P$ values from two-sided unpaired Student's $t$-test. N1ICD $p = 0.0008$, HES1 $p = 0.0009$. **g** Western blot for GUCY1B1 and SYP in CDX17 and CDX17P NE and Non-NE ex vivo cultures ($n = 2$). **h** HEY1, GUCY1B1 and GUCY1A1 mRNA expression (RT-qPCR) of DMSO- or DAPT-treated CDX17P Non-NE cells. $n = 3$, data are represented as mean ± SEM. $P$ values from two-sided unpaired Student's $t$-test. **i** HEY1, GUCY1B1 and GUCY1A1 mRNA expression (RT-qPCR) of empty vector control or N1ICD-overexpressing CDX17P NE cells. $n = 3$, data are represented as mean ± SEM. $P$ values from two-sided unpaired Student's $t$-test. HEY1 $p = 0.0005$. **j** Notch1 ChIP-qPCR in H1048 cells overexpressing N1ICD. qPCR of RBPJ-binding sites in the GUCY1A1 and GUCY1B1 promoter. N1ICD binding to the HES1 promoter (positive control), binding to negative control region (negative control). $n = 3$, data are represented as mean ± SD. $P$ values from two-sided unpaired Student's $t$-test. See also Supplementary Fig. 3.

of the antibody for IHC) (Fig. 5b). Subsequently, sgNTA control cells and sgB1-2 cells were implanted subcutaneously into NSG mice and treated with either vehicle or cisplatin/etoposide. CDX17P sgB1-2 tumours grew significantly slower than control CDX17P sgNTA tumours (Fig. 5c), taking on average 32 days to reach randomization volume (200 mm³), 10 days slower than sgNTA tumours ($p < 0.001$) (Supplementary Fig. 5a). Once randomization volume was reached, tumours exhibited similar growth kinetics (Supplementary Fig. 5b). Consistent with published results[10], cisplatin/etoposide did not elicit a significant response in CDX17P sgNTA tumours; however, chemotherapy treatment slowed CDX17P sgB1-2 tumour growth and significantly increased event-free survival (median 9 and 15 days in vehicle and cisplatin/etoposide cohorts, respectively, $p = 0.004$) (Fig. 5d, e). Lower GUCY1B1 protein expression in sgB1-2 tumours compared to control sgNTA tumours at the conclusion of the study was confirmed by IHC and western blot of tumour lysates (Fig. 5f–h). Staining for cleaved caspase-3 to assess apoptosis revealed a similar induction of apoptosis upon treatment with cisplatin/etoposide in sgNTA and sgB1-2 tumours (Supplementary Fig. 5c). Comparing staining for phospho-histone H3 indicated a lower proliferative rate in cisplatin/etoposide-treated sgB1-2 tumours compared to sgNTA control tumours ($p = 0.041$) (Supplementary Fig. 5d).

In the absence of a potent sGC inhibitor for in vivo use, we tested whether targeting NO production with the NOS inhibitor L-NMMA[38] could potentiate cisplatin/etoposide efficacy in CDX17P. Treatment with cisplatin/etoposide elicited only a minor reduction in tumour growth which was significantly enhanced by L-NMMA co-treatment, resulting in increased event-free survival (Fig. 5i, j, median 13 and 17 days in mice treated with cisplatin/etoposide or cisplatin/etoposide + L-NMMA, respectively, $p = 0.049$).

Taken together, these results further establish sGC signalling as a mediator of acquired chemotherapy resistance in SCLC.

## Discussion
Most biopsies from SCLC patients are taken at the time of diagnosis and the rarity of post relapse tumour samples is a major obstacle to studying the rapid emergence of acquired chemoresistance. CTCs are a readily repeatable liquid biopsy and we took advantage of paired pre-treatment and post-progression patient CDX from six patients with extensive stage SCLC for this study. RNA-seq profiling revealed recurrent upregulation of two subunits of an sGC in 4/6 longitudinal CDX pairs correlating with development of chemoresistance at progression. We discovered a link between chemoresistance and Notch/NO-mediated activation of sGC and PKG. Of note, a recent publication studying metabolic differences between SCLC subtypes discovered a dependency of MYC-driven SCLC on arginine, a precursor for

NO generation, and demonstrated increased dependency on arginine in SCLC cell lines with acquired chemoresistance[6]. The SCLC CDX model we used to study sGCs and acquired resistance to SCLC standard of care chemotherapy in vivo, CDX17P, showed upregulation of MYC compared to this donor's baseline model CDX17[10]. The aforementioned study did not link acquired chemoresistance to NO synthesis via NOS and a relationship between sGC and PKG signalling with development of resistance was not investigated. Nevertheless, the combined results of Chalishazar et al.[6] and our present study warrant further investigation of the relationship between MYC, arginine dependency and increased sGC/PKG signalling. It is also notable that MYC cooperates with NOTCH to drive subtype switching from NE to Non-NE[12,24] and that like MYC[10], GUCY1B1 is expressed in the Non-NE subpopulation of CDX17P, suggesting that the minority population of Non-NE cells in the tumour is responsible for etoposide resistance during disease evolution. An outstanding question is the source of NO within tumours. NO is typically synthesized by endothelial cells and SCLC is a well vascularized tumour[39] as are our CDX models, implicating the vasculature as a possible NO source. Alternatively, we have previously shown that SCLC cells can undergo vasculogenic mimicry, a process whereby cancer cells acquire properties of endothelial cells and form vessel-like structures[40], raising the intriguing possibility that SCLC cells may generate NO themselves, a hypothesis warranting further study.

NO has both pro- and anti-tumourigenic properties, most likely dependent on cancer type and NO levels[22]. Low, physiological levels of NO can induce angiogenesis, metastasis, tumour growth[41], cisplatin resistance[22] and suppress apoptosis in various cell types, including transformed neural cells[42]. The anti-apoptotic effects were linked to stimulation of sGC and subsequent activation of PKG signalling through increased levels of cGMP[43]. Our in vitro studies inhibiting sGC using both chemical and genetic methods indicate that sGC signalling mediates resistance to the topoisomerase II inhibitors etoposide and doxorubicin but not to cisplatin. The majority of patients contributing to the generation of our paired CDX models received a combination of carboplatin/etoposide (CDX3/3P, CDX17/17P, CDX18/18P, CDX42/CDX42P), including the patient donor of CDX17P that was selected for our in vivo studies (Supplementary Table 1). However, the patient donors of CDX8/8P and CDX20/20P did not receive etoposide treatment. Despite this, an upregulation of sGC subunits was observed in their CDX models derived from donor patients at disease progression. Together, these data from the six paired models suggest that evolution of SCLC following chemotherapy (with or without etoposide) results in resistance to topoisomerase II inhibitors and the in vivo study in CDX17P implicates sGC signalling in this resistance. Of note,

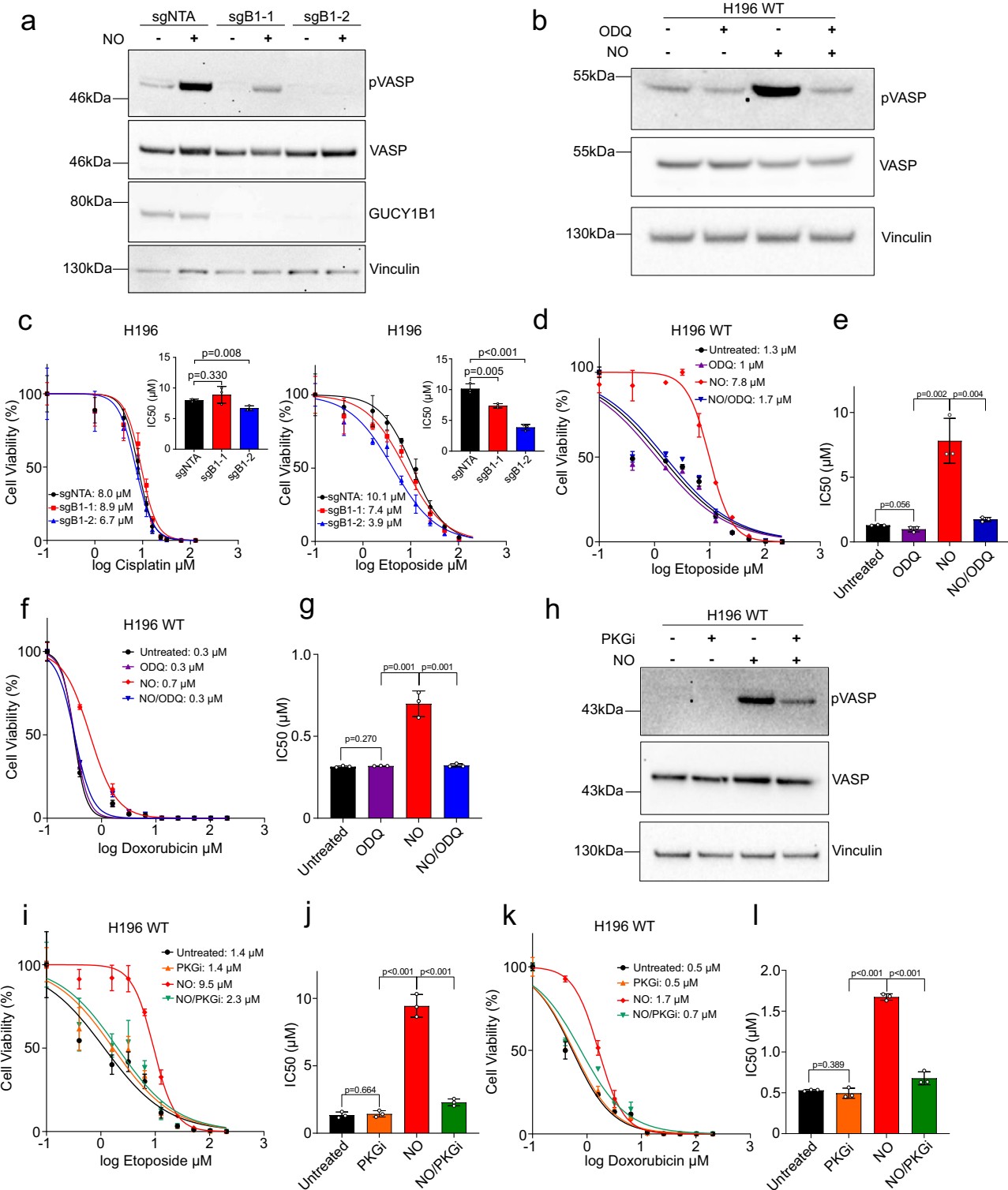

previous studies on acquired resistance in SCLC found that resistance is primarily acquired towards topoisomerase inhibitors and not to cisplatin[4,5].

In the present study, we demonstrate that resistance towards topoisomerase inhibitors in SCLC proceeds through a sGC/PKG axis but without reduction in apoptosis in vivo (measured at single time-point at study conclusion). Although we did not achieve a complete *GUCY1B1* knockout in CDX17P, we were able to significantly increase event-free survival (Fig. 5). Furthermore, our data suggest a role of GUCY1B1 in tumour initiation, since

CDX17P sgB1-2 tumours took significantly longer to reach the specified randomization volume compared to CDX17P sgNTA tumours. These results raise the possibility that SCLC patients whose tumours or CTCs exhibit high levels of GUCY1B1 may benefit from co-targeting sGC or PKG signalling to maintain sensitivity to etoposide. Therapeutic targeting of the sGC pathway has been assessed in NSCLC and breast cancer models. Specific inhibition of PKG-Iα using the inhibitor DT2 increased apoptosis in NSCLC cell lines in vitro and in this context was synergistic with cisplatin treatment[37]. Potent PKG inhibitors for in vivo

**Fig. 4 Impact of sGC signalling on SCLC cell fate and chemotherapy responses in vitro. a** Western blot for pVASP, VASP and GUCY1B1 in H196 sgNTA, sgB1-1, sgB1-2 cells treated with NO and untreated cells ($n = 3$). **b** Western blot for pVASP and VASP of untreated, ODQ (sGC inhibitor), NO and ODQ + NO treated H196 cells ($n = 3$). **c** Viability of H196 sgNTA, sgB1-1, sgB1-2 cells treated with NO and cisplatin or etoposide. Data as mean ± SD. $IC_{50}$ of three independent replicates ($n = 3$). *P* values from two-sided unpaired Student's *t*-test. sgNTA-sgB1-2 etoposide $p = 0.0003$. **d** Viability of untreated, ODQ, NO and ODQ + NO treated H196 cells treated with etoposide. Representative data of three independent replicates are shown and $IC_{50}$ values are indicated. Data as mean ± SD. **e** $IC_{50}$ values of **d** ($n = 3$), data as mean ± SD. *P* values from two-sided unpaired Student's *t*-test. **f** Viability of untreated, ODQ, NO and ODQ + NO treated H196 cells treated with doxorubicin. Representative data of three independent replicates are shown and $IC_{50}$ values are indicated. Data as mean ± SD. **g** $IC_{50}$ values of **f** ($n = 3$), data as mean ± SD. *P* values from two-sided unpaired Student's *t*-test. **h** Western blot for pVASP and VASP of untreated, PKG inhibitor (PKGi), NO and PKG inhibitor + NO treated H196 cells, $n = 3$. **i** Viability of untreated, PKG inhibitor, NO and PKG inhibitor + NO treated H196 cells treated with etoposide. Representative data of three independent replicates are shown and $IC_{50}$ values are indicated. Data as mean ± SD. **j** $IC_{50}$ values of **i** ($n = 3$). *P* values from two-sided unpaired Student's *t*-test. *P* values: PKGi-NO: 0.0001, NO-NO/PKGi: 0.0001. **k** Viability of untreated, PKG inhibitor, NO, and PKG inhibitor + NO treated H196 cells treated with doxorubicin. Representative data of three independent replicates are shown and $IC_{50}$ values are indicated. Data as mean ± SD. **l** $IC_{50}$ values of **k** ($n = 3$). *P* values from two-sided unpaired Student's *t*-test. *P* values: PKGi-NO: 0.0001, NO-NO/PKGi: 0.0001. See also Supplementary Fig. 4.

studies and clinical trials are currently unavailable[44]. An alternative strategy could be to target NOS, which is responsible for the generation of NO and consequently the activation of sGC and PKG. Consistent with published data demonstrating that a NOS inhibitor decreased *KRAS*-driven NSCLC tumour growth and synergized with chemotherapy[45], combination of L-NMMA with cisplatin/etoposide delayed tumour growth and significantly increased event-free survival in CDX17P. Although NOS inhibitors have not been tested in a cancer setting, their safety and efficacy has been tested in a clinical trial to restore mean arterial pressure (NCT00835224), suggesting targeting NOS is a worthy therapeutic avenue to pursue.

There are multiple, valid approaches to uncover mechanisms of acquired resistance in SCLC. A commonly used approach is to repeatedly treat mouse models (cell line xenografts or PDX) with increasing drug concentrations until resistance emerges. Transcriptional downregulation of *SLFN11* has been identified as one mechanism of SCLC acquired chemoresistance using this drug re-challenge strategy to study ten PDX mouse models driven to chemotherapy resistance[5]. However, comparing SLFN11 transcript and protein expression levels in CDX and PDX models, Drapkin et al.[7] did not identify a difference between models derived from untreated and previously treated patients and we could also not observe decreased *SLFN11* in our progression CDX. Another approach is to interrogate patient biospecimens to identify alterations that correlate with acquired resistance, although this approach is limited by scarce availability of paired pre- and post-treatment biopsies and an inability to function test candidate mechanisms in the clinical material. Wnt-pathway genes[4] have been implicated in SCLC resistance via WES performed on paired tumour samples from 12 SCLC patients; we did not observe these changes in our CDX study.

Our study for the first time employs transcriptomic analysis on six paired SCLC patient-derived baseline and progression models to uncover mechanisms of acquired chemoresistance, where in contrast to previous mouse modelling studies, acquired resistance occurred in the patient.

These disparities in studies on acquired resistance mechanisms could be explained by the different model systems. Furthermore, we did not detect GUCY1B1 and GUCY1A1 upregulation in all six CDX progression models. We did not expect sGC signalling to be the sole reason for chemoresistance given the high degree of ITH in relapsed SCLC (demonstrated by single cell RNA-seq analysis in eight CDX models[11]) that infers likely existence of multiple chemoresistance mechanisms. We conclude that whichever approach is used to discover mechanisms of acquired chemoresistance, and we favour the paired CDX approach, in vivo validation followed by clinical testing are the required next steps.

In summary, by generating paired pre-treatment and post-progression CDX models from patients with SCLC, we were able to show a recurrent upregulation of sGCs during SCLC progression that correlated with acquired chemotherapy resistance in vivo. sGC subunit expression was regulated by Notch signalling and activated by NO and sGC driven chemoresistance was mediated through PKG. This study reveals a potential vulnerability in relapsing SCLC. Analysis of sGC subunits in SCLC biopsies or in readily accessed CTCs could provide a predictive biomarker of response to sGC pathway inhibition in a disease for which novel treatment strategies are urgently needed.

## Methods

**In vivo animal studies.** For CDX model generation and studies testing efficacy of cisplatin/etoposide in sgNTA and sgB1-2 CDX17P, 8–16-week-old female non-obese diabetic (NOD) severe combined immunodeficient (SCID) interleukin-2 receptor γ-deficient (NSG) mice (Charles River) were used. For L-NMMA efficacy studies, guided by a previous study testing L-NMMA and to increase tolerability to the compound[38], 9–11-week-old female SCID Beige mice (Envigo) were used. Furthermore, all mice were drug/test naive, did not undergo previous procedures and were housed in individually vented caging systems in a 12-h light/12-h dark environment and maintained at ambient temperature and humidity, and any cell implants and dosing were carried out in the morning on a laminar air flow bench and mice placed back in their home cages. For CDX model generation, 10 ml of peripheral blood was collected from SCLC patients adhering to the ethically approved CHEMORES protocol (07/H1014/96). Information about the patients contributing to CDX model generation can be found in Simpson et al.[10]. Subsequently, blood was processed using the RosetteSep CTC Enrichment Cocktail (Stemcell Technologies, 15167), cells resuspended in 100 μl of a 1:1 mixture of HITES medium (RPMI 1640 supplemented with 50 μg/ml insulin (Merck, I9278), 100 μg/ml transferrin (Merck, T8158), 100 nM hydrocortisone (Merck, H0888), 300 nM sodium selenite (Merck, S5261-100), 100 nM β-estradiol (Merck, E2758)) with Matrigel (BD Biosciences, 354234) and implanted subcutaneously into the flank of NSG mice. When CDX tumours reached 600 mm³ (passage one, p1), they were dissected into 3 × 3 mm³ fragments, which were re-implanted into flanks of five NSG mice (p2). Finally, p2 tumours were disaggregated and re-implanted into five NSG mice (p3). To disaggregate tumours for CDX ex vivo cultures and to passage p2 tumours, a gentleMACS Octo dissociator was used (Miltenyi Biotec, 130-095-937). To assess sensitivity of CDX models to treatment with cisplatin and etoposide, 15 female NSG mice were implanted with CDX17P sgNTA or CDX17P sgB1-2 cells with the expectation that five tumours would not grow successfully, leaving five animals per treatment group (vehicle vs. cisplatin and etoposide). Mice were randomized deterministically at 150–250 mm³ by assignment to vehicle or cisplatin and etoposide treatment groups, by evenly distributing initial tumour volume sizes. Cohort size was guided by a study by Murphy et al.[46], demonstrating that a cohort size as few as one mouse can predict treatment response. In order to minimize potential confounding factors, tumour measurements were performed by members of the laboratory not directly involved in the project and animals belonging to different treatment groups were housed in separate cages. Group allocations were known during allocation, conducting, outcome assessment and data analysis of experiment and no animals were excluded throughout data analysis. 5 mg/kg cisplatin dosed at 5 ml/kg (Christie Pharmacy Ltd), 8 mg/kg etoposide dosed at 5 ml/kg (Sigma, 33419-42-0) in *N*-methyl-2-pyrrolidone (NMP) and citric acid, and vehicle compound (0.9% saline solution and NMP, respectively) was administered by intraperitoneal injection on day 1 and on days 1, 2, and 3, respectively, or corresponding vehicle control. Mice were monitored at least twice a week by caliper until four times initial tumour volume was reached (4× ITV) or

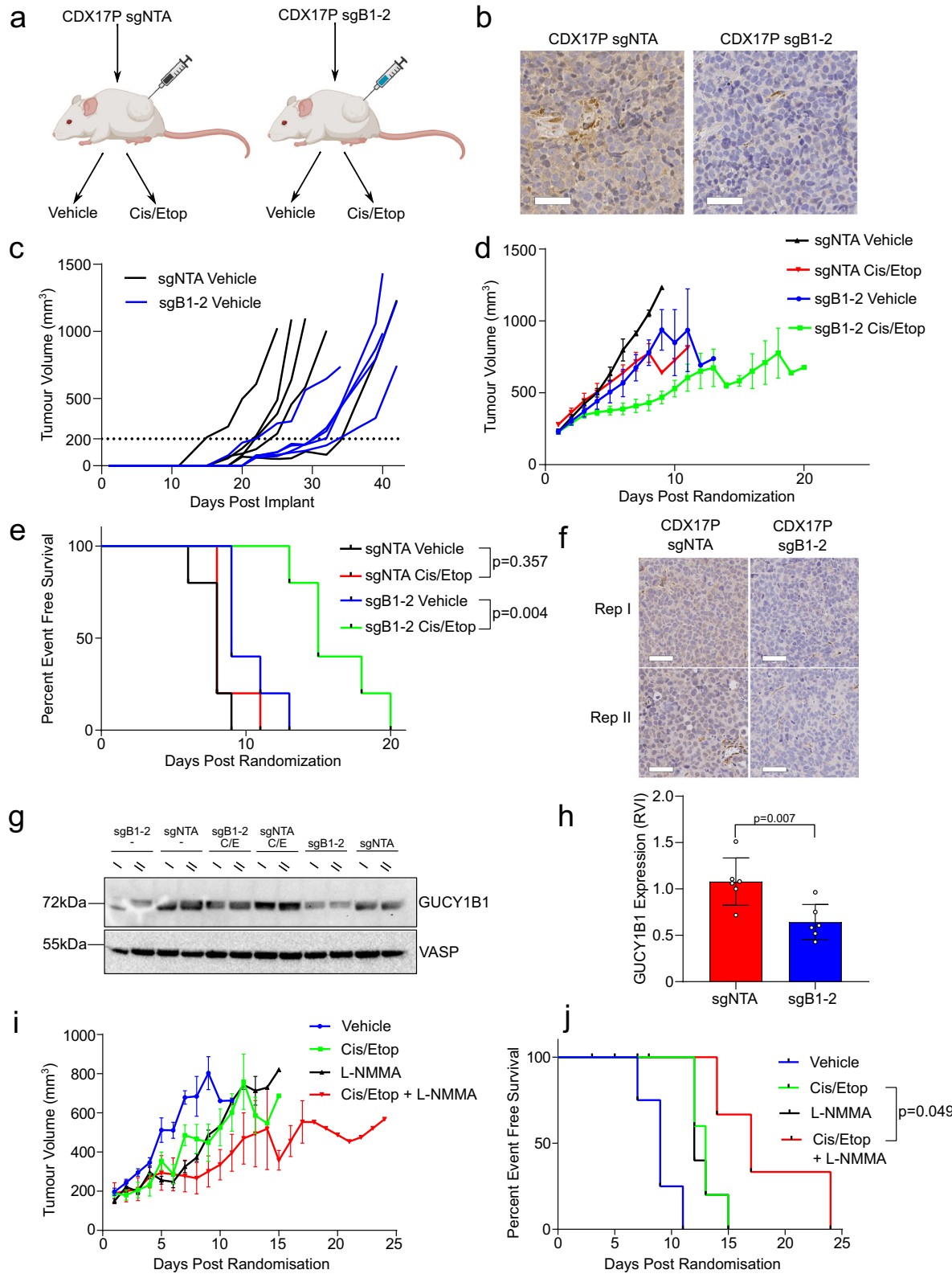

until animal health deteriorated. To test whether targeting NO production with the NOS inhibitor L-NMMA could potentiate cisplatin/etoposide efficacy in CDX17P, 25 SCID Beige mice were implanted subcutaneously with CDX17P with the expectation that five mice would not grow tumours, leaving five mice per treatment group. Mice were treated with cisplatin/etoposide as described above, 200 mg/kg L-NMMA dosed at 5 ml/kg (Hello Bio, made up in sterile water) on days −1 to 0 and 2 to 5 and with 400 mg/kg L-NMMA dosed at 10 ml/kg on day 1 by oral gavage. Furthermore, in order to alleviate side effects of increased mean systolic pressure caused by treatment with L-NMMA, mice were treated with 10 mg/kg of the anti-

hypertensive amlodipine (Christie Pharmacy Ltd) by oral gavage on days of L-NMMA treatment[38]. Mice for which dosing schedules could not be completed due to a deterioration in clinical condition or body weight loss were excluded throughout data analysis. All procedures were performed in accordance with the Home Office Regulations (UK) and the UK Coordinating Committee on Cancer Research guidelines, by approved protocols (Home Office Project license nos. 40-3306/70-8252/P3ED48266), and the Cancer Research UK Manchester Institute Animal Welfare and Ethical Review Advisory Body. CDX samples are available on reasonable request via discussion with the corresponding author.

**Fig. 5 Loss of sGC signalling sensitizes CDX17P to cisplatin/etoposide in vivo. a** Schematic of our strategy to study chemotherapy responses in vivo. **b** IHC for GUCY1B1 expression (brown stain) of CDX17P sgNTA and sgB1-2 tumours to demonstrate maintenance of reduced GUCY1B1 expression after in vivo passage. Scale bar set to 50 μm and equivalent throughout panels. Whole tumours were analysed and representative areas are shown. **c** Individual tumour volumes of vehicle-treated mice implanted with CDX17P sgNTA or CDX17P sgB1-2 cells. Data are from five animals per experimental arm. **d** Mice implanted with CDX17P sgNTA or CDX17P sgB1-2 cells were treated with a combination of 5 mg/kg cisplatin and 8 mg/kg etoposide or vehicle treated. Data from five animals per experimental arm and represented as mean ± SEM. **e** Kaplan–Meier survival curve comparing vehicle and cisplatin/etoposide-treated mice until the tumour reaches 4× initial tumour volume (ITV). $P$ values from two-sided log-rank test, chi square = 8.221, d.f. = 1. **f** IHC of CDX17P sgNTA and sgB1-2 tumours for GUCY1B1 (brown stain). Scale bar set to 50 μm and equivalent throughout panels. Whole tumours were analysed and representative areas are shown. Two representative replicates of CDX17P sgNTA and sgB1-2 tumours are shown. **g** Western blot for GUCY1B1 and VASP on tumour lysates derived from two replicates per experimental arm. (−): Vehicle-arm, (C/E): cisplatin/etoposide-arm, lanes 9-12: non-randomized mice. VASP served as a loading control. **h** Quantification of GUCY1B1 expression (RVI) of **g**, normalized to VASP expression. Six replicates for sgNTA and sgB1-2 cohorts were analysed ($n = 6$), data are represented as mean ± SD. $P$ values from two-sided unpaired Student's $t$-test. $t = 3.367$, d.f. = 10. **i** Mice implanted with CDX17P were treated with a combination of cisplatin/etoposide, L-NMMA, cisplatin/etoposide + L-NMMA or vehicle treated. Data are from three to five animals per experimental arm and represented as mean ± SEM. **j** Kaplan–Meier survival curve comparing percent event-free survival of mice until the tumour volume reaches 4× initial tumour volume (ITV). $P$ values from two-sided log-rank test, chi square = 3.875, d.f. = 1. See also Supplementary Fig. 5.

**Cell lines and primary cultures**. SCLC cell lines NCI-H196 (source: male; RRID:CVCL_1509), NCI-H524 (source: male; RRID:CVCL_1568), NCI-H1339 (source: female; RRID:CVCL_A472), NCI-H82 (source: male; RRID:CVCL_1591), and NCI-H446 (source: male; RRID:CVCL_1562) were maintained in RPMI 1640 (Thermo Fisher Scientific, 21875091) supplemented with 10% FBS (Labtech, FCS-SA) and 1× penicillin–streptomycin (Sigma-Aldrich, P0781) in a humidity controlled environment (37 °C, 5% $CO_2$). NCI-H1048 (source: female; RRID: CVCL_1453) cell line was maintained in HITES media supplemented with 5 μM of Y-27632 (Tocris, 1254-10) with addition of 5% FBS in a humidity controlled environment (37 °C, 5% $CO_2$). Cell lines were obtained from the American Type Culture Collection (ATCC), tested negative for mycoplasma using a Venor®GeM-qEP Mycoplasma Detection Kit (Cambio, 11-9250) run on a QuantStudio 5 Real-Time PCR System (Thermo Fisher Scientific). Cell lines were confirmed by STR profiling using the Promega PowerPlex 21 kit (Promega, DC8902) and analysed using genemapper5 software and an in-house database for comparisons/matching.

To obtain primary CDX ex vivo cultures, CDX tumours were disaggregated at passage 3 (p3) using a gentleMACS Octo dissociator. After disaggregation, cells were grown in HITES media with addition of 2.5% FBS in a humidity controlled environment (37 °C, 5% $CO_2$).

**SCLC TMAs**. Material for the SCLC TMA was sourced through the approved Manchester Cancer Research Centre Biobank (MCRC), Project Reference 10_FIBL_01. Patients were identified in collaboration with the clinical accredited pathology department at Wythenshawe Hospital. A pathology review was conducted by a lead pathologist on those patients who had surgically resected and histologically confirmed SCLC (diagnosed 1993-2005) stored in the department which were initially used for diagnostic requirements but became surplus to diagnostic purpose. As these samples were an existing holding held prior to the HTA commencement date of 1 September 2006, it negated the requirement for patient consent. Specimens were processed to formalin-fixed paraffin-embedded (FFPE) blocks in line with pathology department-approved SOPs. The TMAs were constructed in accordance to MCRC Biobank-approved SOPs. For those patient samples that contributed to the CDX generation, patients gave written informed consent to donate blood samples, pre, during and post-treatment under the ethically approved ChemoRes (CHEMOtherapy RESistance) study, ethics reference—07/H1014/96 approved by the North West Greater Manchester West Research Ethics Committee. The focus of this study is to investigate blood borne biomarkers for disease resistance in lung cancer patients.

**RNA sequencing**. RNA was extracted from RNAlater (Sigma-Aldrich, R0901)-treated tumours derived from three independent mice per CDX model by homogenizing tissue in Fastprep tubes with matrix A (MP Biomedical, 116910500) containing RLT buffer using a TissueLyserLT (Qiagen, 69980). RNA was extracted using the RNeasy mini kit (Qiagen, 74104) and a DNase (Qiagen, 79254) digest performed to remove leftover DNA. RNA was quantified using a Qubit™ RNA HS Assay kit (Thermo Fisher Scientific, Q32855) and RNA with an integrity number >8 determined using a Bioanalyzer RNA 6000 Nano assay (Agilent, 5067-1511) was taken forward to generate libraries. Indexed PolyA libraries were prepared using 200 ng of total RNA and 14 cycles of amplification with the SureSelect Strand Specific RNA-seq Library Preparation kit for Illumina Sequencing (Agilent, G9691B). Library quality was checked using the Agilent Bioanalyzer. Libraries were quantified by qPCR using the Kapa Library Quantification Kit for Illumina (Roche, 07960336001). Paired-end 2 × 75 bp sequencing was carried out by clustering 1.8–2.0 pm of the pooled libraries on the NextSeq 500 sequencer (Illumina Inc.). RNA-seq data were aligned to *Homo sapiens* GRCh38 and Mouse GRCm38 assembly (Ensembl release 99) using STAR version 2.6.1d[47] as part of the nf-core RNA-seq pipeline[48]. These data were filtered using the bamcmp algorithm version 2.0 (ref. [49]) to remove any mouse contamination reads. The counts matrix was

generated using the filtered reads and the Rsubread package version 2.0.1. Differential expression analysis was conducted using DESeq2 (version 1.26.0)[50] and model matrix accounted for paired testing. The log2 fold change values were shrunk via the apeglm algorithm version 1.12.0 (ref. [51]) within DESeq2, and volcano plots generated using Enhanced Volcano version 1.8.0 (ref. [52]). Gene set enrichment analysis was performed using generally applicable gene set enrichment for pathway analysis (GAGE) (version 2.36)[53]. Sample similarity (Supplementary Fig. 2b) is based on a Euclidian distance metric between VST normalized gene expression profiles across all samples and clustered using the complete linkage estimator.

**Whole-exome sequencing**. Genomic DNA was extracted from flash-frozen CDX tumours using the DNeasy Blood and Tissue kit (Qiagen, 69504) by incubating tissue overnight (O/N) in buffer ATL with proteinase K at 56 °C. Genomic DNA was quantified using a Qubit™ dsDNA HS Assay Kit (Thermo Fisher Scientific, Q32854) and run on the Agilent Tapestation Genomic DNA assay (Agilent, 5067-5365). Genomic DNA was sheared on the Bioruptor Pico sonication system (Diagenode, B01080010) to an average size of 150–200 bp. Libraries were prepared using 200 ng of sheared genomic DNA, 10 cycles of pre-capture PCR and 11 cycles of post-capture PCR, with the SureSelectXT Target Enrichment System for Illumina Paired-End Sequencing (Agilent, G9641B). Low input samples were prepared using 8–200 ng of sheared genomic DNA, 8–11 cycles of pre-capture PCR and 9 cycles of post-capture PCR, with the SureSelectXT Low Input Target Enrichment System for Illumina Paired-End Sequencing (Agilent, G9703A). SureSelectXT Human All Exon V6 Capture Library (Agilent, 5190-8865) was used for all samples. Library quality was checked using the Agilent Bioanalyzer. Libraries were quantified by qPCR using the KAPA Library Quantification Kit for Illumina (Roche, 07960336001). Paired end 2 × 101 bp sequencing was carried out by clustering 14 pM of the pooled libraries on the HiSeq 2500 sequencer in High Throughput mode with V3 chemistry (Illumina, Inc.). Adapter sequences were removed from the reads using Cutadapt version 2.10 (ref. [54]). Alignment of WES data to Human reference genome GRCh38 and Mouse reference genome GRCm38 was performed using bwa-mem version 0.7.17 (ref. [55]). Reads originating from potential mouse contamination were removed using bamcmp version 2.0, an algorithm to distinguish human and mouse reads[49]. Picard version 2.19.0 (ref. [56]) and GATK tools version 4.1.7 (ref. [57]) was used for deduplication, realignment, and recalibration of aligned data. Mutect2 version 4.1.7 (ref. [58]) was used to call somatic mutations (TP53 mutations were also present in the germline of patient 18 and 20 and RB1 mutations were present in the germline of patient 20) and VEP version 99 (ref. [59]) was used to annotate mutation calls. CN data were generated using CNVkit version 0.9.3 (ref. [60]). Cancer mutational signatures were identified from the variant calls using SigsPack in R version 1.4.0 (ref. [61]).

**Immunohistochemistry**. IHC was performed on FFPE cores of SCLC tumours or CDX tumours, and staining was done on 4 μm sections using recombinant anti-GUCY1B1 antibody (Abcam, ab154841) at 6.76 μg/ml, pHH3 (Millipore, 06-570) at 0.4 μg/ml, and cCas3 (Cell Signaling Technology, 9661) at 0.2595 μg/ml diluted in Bond Primary Antibody Diluent (Leica Biosystems, AR9352). Heat-induced epitope retrieval (HIER) was performed using BOND Epitope Retrieval Solution 1 (Leica Biosystems, AR9961) for 20 min and staining was carried out using the Leica Bond Max Platform performing standard protocol F with Bond Polymer Refine Detection (Leica Biosystems, DS9800). Scanning of the stained slides was performed using a Leica SCN400. Digitally scanned slides were analysed using HALO software v2.3 (Indica Labs).Tumour regions were defined within each sample using a machine learning classifier. Tumour cells were detected and classified as positive or negative based on IHC thresholds using the Area Quantification algorithm. Expression level was quantified as percentage positive tumour within the sample.

**Co-immunofluorescence**. FFPE CDX NE and Non-NE cells were cut as 4 µm sections and stained by IHC for REST monoclonal antibody (CL0381) 1:150 (Thermo Fisher Scientific, MA5-24606, RRID:AB_2637221) and SYP (Leica Biosystems, PA0299) on a Leica Bond Max Platform using standard protocol F with Bond Polymer Refine Detection kit (DS9800). GUCY1B1 and HES1 co-IF was performed on a Leica Bond Rx Platform using the PerkinElmer Opal 4-Colour Automation IHC Kit (NEL800001KT). Tissue sections were blocked with 3% hydrogen peroxide (Sigma-Aldrich, H1009) for 10 min to block endogenous peroxidase activity, followed by 10% casein solution (Vector Laboratories, SP-5020) for 10 min to block non-specific antibody binding. Slides were stained with GUCY1B1 primary antibody 1:100 (Abcam, 154841) followed by DAKO envision + system horseradish peroxidase-coupled secondary antibody (DAKO, K4003, RRID:AB_2630375) for 30 min, followed by incubation with OPAL Tyramide-fluorophore (OPAL570, 1:200) for 10 min. Slides were heat-inactivated following the tyramide-fluorophore incubation step, then blocked and probed with HES1 primary antibody 1:100 (Cell Signalling Technologies,11988, RRID:AB_2728766), followed by DAKO envision+ system HRP-conjugated secondary antibody (DAKO, K4003) for 30 min, followed by incubation with OPAL tyramide-fluorophore (OPAL650, 1:200) for 10 min. Cells were stained with DAPI (1:1000) for 15 min and slides were mounted in Molecular Probes ProLong Gold Antifade Mountant (Thermo Fisher Scientific, P36934). Slides were digitally scanned using an Olympus VS120.

**Western blotting**. Cell lysates were prepared from cell pellets or flash-frozen tissue using the CST cell lysis buffer (Cell Signaling Technology, 9803S) in the presence of Protease Inhibitor Cocktail (Merck, P8340) and Phosphatase Inhibitor Cocktail II (Merck, P0044) and III (Merck, P5726). For extraction from flash-frozen CDX tumours, tissue was homogenized in Fastprep tubes with matrix A using a TissueLyserLT in ice-cold lysis buffer. Crude lysates were clarified by centrifugation at 18,800$g$ for 15 min in a refrigerated bench top centrifuge (Eppendorf 5417 R). A BCA protein assay reagent kit (Thermo Fisher Scientific, 23225) was used to determine protein concentrations and lysates resuspended in 10× NuPAGE sample reducing agent (Thermo Fisher Scientific, NP0009) and 4× NuPAGE LDS sample buffer (Thermo Fisher Scientific, NP0007). Protein lysates were resolved on a NuPAGE 4–12% Bis-Tris 1.0 mm gel (Thermo Fisher Scientific, NP0322BOX) using MOPS SDS running buffer (Thermo Fisher Scientific, NP0001). Western blots were transferred onto PVDF membranes (Thermo Fisher Scientific, 10617354) in NuPAGE transfer buffer (Thermo Fisher Scientific, NP00061) and membranes were blocked in 5% milk TBS blocking buffer supplemented with 0.2% Tween (Merck, T2700) (TBST) for 1 h at room temperature with agitation. Subsequently membranes were incubated with corresponding primary antibodies in 5% milk TBST overnight at 4 °C (Rabbit recombinant anti-GUCY1B1 antibody 1:1000 (Abcam, 154841), Rabbit phospho-VASP (Ser239) antibody 1:1000 (Cell Signaling Technology, 3114, RRID:AB_2213396), Rabbit anti-VASP Antibody 1:5000 (Bethyl Laboratories, A304-769A-M, RRID:AB_2782159), Rabbit anti-Synaptophysin antibody 1:20,000 (Abcam, ab32127, RRID:AB_2286949), Rabbit HES1 (D6P2U) mAb 1:500 (Cell Signaling Technology, 11988, RRID:AB_2728766), Rabbit GAPDH (14C10) mAb 1:5000 (Cell Signaling Technology, 2118, RRID:AB_561053), Rabbit Notch1 Antibody 1:500 (Bethyl Laboratories, A301-895A, RRID:AB_1524102), Rabbit Histone H3 Antibody 1:5000 (Cell Signaling Technology, 9715, RRID:AB_331563), Rabbit α-Tubulin Antibody 1:5000 (Cell Signaling Technology, 2144, RRID:AB_2210548), Rabbit Vinculin Antibody 1:20,000 (Abcam, ab129002, RRID:AB_11144129)) and the appropriate horseradish peroxidase-coupled secondary IgG 1:10,000 (Agilent, P044801-2, RRID:AB_2617138) in 5% milk TBST for 1 h at room temperature. Membranes were developed using the Supersignal West Femto Chemiluminescent Substrate (Thermo Fisher Scientific, 34094) and the BioRad ChemiDoc XRS + System (BioRad, 1708265). Images were analysed using BioRad software Image Lab 3.0.1. Unprocessed scans of blots are available in the Source Data file.

**Plasmid generation and lentiviral production**. To generate CRISPR/Cas9 knockout derivatives, sgRNAs were designed using CHOPCHOP v3 (ref. [62]), sgNotch1-1 sequences derived from Jiao et al.[63], and sgNTA sequences were derived from Joung et al.[64] and inserted into the lentiCRISPR v2 plasmid (see Supplementary Table 2 for sgRNA sequences). lentiCRISPR v2 was a gift from Feng Zhang (Addgene plasmid # 52961; RRID: Addgene_52961)[65]. Five micrograms of lentiCRISPRv2 was digested with FastDigest Esp3I (Thermo Fisher Scientific, FD0454) for 30 min at 37 °C in the presence of FastAP (Thermo Fisher Scientific, EF0654) and 10mM DTT (Merck, 10197777001) and afterwards run on a 1% agarose gel, followed by gel purification using a QIAquick Gel extraction kit (Qiagen, 28706). Oligos (100 µM) were phosphorylated and annealed using a T4 PNK (NEB, M0201S) at the following conditions: 37 °C for 30 min and 95 °C for 5 min ramping down to 25 °C at 5 °C/min. Afterwards, ligation was performed using 50 ng of digested lentiCRISPRv2 and a 1:200 dilution of phosphorylated and annealed oligos and a Quick Ligase (NEB, M2200S). Transformations were performed using 5 µl of ligated vector and 50 µl of NEB 5alpha competent *E. coli* (High Efficiency) (NEB, C2987U). Cells were incubated on ice for 30 min, followed by a heat shock at 42 °C for 30 s, incubation on ice for 5 min, 900 µl SOC added and cells were recovered by shaking cells at 37 °C for 1 h. Afterwards, cells were plated on 10 cm diameter agar plates containing LB and 100 µg/ml ampicillin (Merck, A9518). To isolate the plasmids, a QIAprep Spin Miniprep Kit (Qiagen,

27106) was used. To verify successful cloning of the sgRNAs into the vector, samples were sent for Sanger sequencing. Finally, plasmid preparation was done performing Maxi Preps using the NucleoBond® Xtra Maxi EF kit (Macherey-Nagel, 740424.50). pLIX-hN1ICD was a gift from Julien Sage (Addgene plasmid# 91897; RRID: Addgene_91897). pLIX_403 was a gift from David Root (Addgene plasmid# 41395; RRID: Addgene_41395). For lentivirus production, LentiX cells were transfected at 70% confluency in a 10 cm dish with 8.5 µg transfer plasmid, 3.4 µg pMDL, 1.7 µg VSVG, and 3.4 µg REV using FuGENE (Promega, E2311). pMDLg/pRRE was a gift from Didier Trono (Addgene plasmid# 12251; RRID:Addgene_12251)[66], pCMV-VSV-G was a gift from Bob Weinberg (Addgene plasmid# 8454; RRID:Addgene_8454)[67], pRSV-Rev was a gift from Didier Trono (Addgene plasmid# 12253; RRID:Addgene_12253)[66]. On the following day, medium was replaced with DMEM (10% FBS, glutamine). Forty-eight hours post transfection, virus was harvested, centrifuged for 5 min at 300$g$ and supernatant filtered through a 0.45 µm acrodisc syringe filter (VWR, 514-4101). Afterwards, $5 \times 10^6$ CDX cells were infected with 1 ml of virus performing spin infection by spinning cells in a single well of a six-well plate at 840$g$ for 45 min at 37 °C in the presence of 6 µg/ml polybrene (Merck, TR-1003-G). Alternatively, adherent cell lines were infected in a 10 cm dish at 70% confluency by applying 1 ml of virus in the presence of polybrene. One day later, virus containing medium was changed with fresh medium. Forty-eight hours post infection, cells were selected daily for 7 days with puromycin (Merck, P8833).

**Viability assays and inhibitors/activators**. SCLC cell lines NCI-H196, NCI-H524, NCI-H1339, NCI-H82, and NCI-H446 were maintained in RPMI 1640 (Thermo Fisher Scientific, 21875091) supplemented with 10% FBS (Labtech, FCS-SA) and 1× penicillin–streptomycin (Sigma-Aldrich, P0781) at 37 °C, 5% CO2. NCI-H1048 cell line was maintained in HITES media supplemented with 5 µM of Y-27632 (Tocris, 1254-10) with addition of 5% FBS at 37 °C, 5% CO2. Cells were treated with 100 µM DETA NONOate (Cayman Chemical, CAY82120) formulated in water, 20 µM PKG inhibitor (Cayman Chemical, CAY15995) formulated in DMSO, 10 µM BAY 41-2272 (Enzo Life Sciences, ALX-420-030-M005) formulated in DMSO, 10 µM DAPT (Merck, D5942) formulated in DMSO, 5 µM DBZ (Tocris, 4489/10), 20 µM ODQ (Cayman Chemical, CAY81410) formulated in DMSO. Cell viability assays were performed by seeding $3–10 \times 10^4$ cells in 100 µl/well in black 96-well plates (Greiner Bio-One, 655090) and drugs were added one day post seeding at increasing concentrations to a final volume of 200 µl/well. Cell viability was assessed by performing Cell-Titer Glo 3D assays (Promega, G9683) 7 days after treatment with cisplatin, etoposide (Merck, E1383), and doxorubicin (Cayman Chemical, CAY15007) in 96-well plates and analysed on a FLUOstar Omega plate reader (BGM Labtech) and software version 5.11 R3. Cell lines were obtained from the ATCC, tested negative for mycoplasma using a Venor®GeM-qEP Mycoplasma Detection Kit (Cambio, 11-9250) run on a QuantStudio 5 Real-Time PCR System (Thermo Fisher Scientific). Cell lines were confirmed by STR profiling using the Promega PowerPlex 21 kit (Promega, DC8902) and analysed using genemapper5 software and an in-house database for comparisons/matching.

**RT-qPCR and ChIP**. RNA from cultured cells was isolated using the RNeasy mini kit and a DNase digest was performed to remove leftover DNA, and cDNA was made using the High- Capacity cDNA Reverse Transcription kit (Thermo Fisher Scientific, 4374967) in a ProFlex™ 2 × 96-well PCR System (Thermo Fisher Scientific, 4484076) and the following conditions: 25 °C for 10 min, 37 °C for 120 min, 85 °C for 5 min. RT-qPCR was performed using the TaqMan Gene Expression Master Mix (Thermo Fisher Scientific, 4369016) in a LightCycler® 96 Instrument software version 1.1.0.1320 (Roche, 05815916001) and the following conditions: 50 °C for 2 min, 95 °C for 10 min, followed by 45 cycles with 95 °C for 15 s and 60 °C for 60 s. All primers were designed using the Thermo Fisher Scientific website (GUCY1A1: Hs01015574_m1, GUCY1B1: Hs00168336_m1, HEY1: Hs05047713_s1, B2M: Hs00187842_m1). Data were normalized to B2M. ChIP protocol was adapted from Lee et al.[68]. Dual-cross-linking of cells was performed using ChIP crosslink gold (Diagenode, C01019027) as well as formaldehyde (Merck, F8775), followed by sonication for 30 s pulses 12 times at 4 °C. ChIP was performed using the following antibodies: 5 µg Rabbit Notch1 antibody (Bethyl Laboratories, A301-895A, RRID:AB_1524102), 5 µg Rabbit (DA1E) mAb IgG XP Isotype Control (Cell Signaling Technology, 3900). RT-qPCR was performed using the SensiFAS SYBR Hi-ROX Kit (Meridian Bioscience, BIO-92005) in a Light-Cycler 96 Instrument (Roche, 05815916001) and the following conditions: 95 °C for 3 min, followed by 40 cycles with 95 °C for 10 s and 62 °C for 15 s, and one cycle with 95 °C for 10 s, 65 °C for 60 s, and with 97 °C for 1 s. Primers in GUCY1A1 and GUCY1B1 promoter regions were derived from Chang et al.[26], HES1 qPCR primers were derived from Xu et al.[69], and negative control primers were adapted from Cheng et al.[70] (see Supplementary Table 2 for primer sequences).

**IncuCyte proliferation assay**. For proliferation assays, cells were cultured O/N in the presence of 100 µM DETA NONOate and on the following day 4000 cells/well seeded into 96-well plate in the presence of 100 µM DETA NONOate. Cell proliferation was monitored by analysing the occupied area over a period of 7 days using an IncuCyte ZOOM System Sartorius and software version 2016A and scans were performed every 12 h.

**Migration/invasion assay**. For migration and invasion assays, cells were serum starved O/N in the presence of 100 μM DETA NONOate (Cayman Chemical, CAY82120). On the following day, 50,000 cells were seeded on a Biocoat Tumor Invasion System (Corning, 354166) or transwell inserts (Corning, 3422) in RPMI 1640 without serum in the presence of DETA NONOate and assays performed O/N. Receiver wells contained RPMI with or without FBS supplemented with DETA NONOate. For migration assays, individual transwells were incubated in 0.5% crystal violet (Sigma-Aldrich, C6158) for 20 min at RT, excess crystal violet washed off and cells removed from the top of the insert. Pictures of individual transwell were taken on a Primovert microscope (Zeiss) on a ×4 magnification, signal quantified using Fiji (version 1.52f)[71] and the fold change of migration compared to sgNTA control cells was calculated. For invasion assays, transwells were inserted into 0.25 μg/ml calcein AM (Corning, 354216) and incubated for 1 h at 37 °C (5% $CO_2$), fluorescence detected using a FLUOstar Omega plate reader (BGM Labtech) software version 5.11 R3 at 485 and 520 nm, and fold invasion to sgNTA control cells was calculated. Additionally, images were taken on an ALS CellCelector (Automated Lab Solutions).

**Quantification and statistical analysis**. All tests used for statistical analysis are detailed in the figure legends of corresponding figures and GraphPad Prism Version 8.2.0 was used for all statistical analysis except for differential expression analysis of RNA-seq data, WES data, and TMA survival analysis, which have been done using R version 3.6.1, and power analysis was performed in SAS software version 9.4. Normality was confirmed performing Shapiro–Wilk test or D'Agostino-Pearson test. Survival analysis of 33 treatment-naive SCLC patients was performed on % positive staining. A Cox proportional hazard regression of the % positive tumour scores was performed, respecting the linearity and proportionality assumptions. Prior to constructing Kaplan–Meier curves the % positive tumour scores were dichotomized using a score of 12 as cut-point. This optimal cut-point was calculated by maximizing the Youden index of the ROC curve and confirmed by the maximally selected rank statistics method. Kaplan–Meier (log rank) analysis was subsequently performed to estimate the median survival in each group. To compare GUCY1B1 expression with stage of disease ($n = 37$ for limited and $n = 7$ for extensive stage patients) and with response of treatment (response summarizes partial and complete response ($n = 26$), no response stable and progressive disease ($n = 8$)), unpaired Student's $t$-test was performed following Shapiro–Wilk test to confirm normality. Power analysis was based on the Cox proportional regression assuming complete event probability, 80% power, 0.05 significance and the extrapolation of the results of this pilot study (HR = 1.02, $sd_{TMA} = 11.66$). d.f.- and $t$-values for individual experiments can be found in the Source Data file.

**Reporting summary**. Further information on research design is available in the Nature Research Reporting Summary linked to this article.

## Data availability

The RNA-seq data generated in this study have been deposited in the EMBL-EBI ArrayExpress database under accession code E-MTAB-8465, titled 'RNA of small cell lung cancer circulating tumour cells derived explants'. The WES data generated in this study have been deposited under accession code E-MTAB-10880, titled 'Soluble guanylate cyclase signalling mediates etoposide resistance in progressing small cell lung cancer'. The GenBank accession codes to Human reference genome GRCh38 is GCA_000001405.15 (https://www.ncbi.nlm.nih.gov/assembly/GCF_000001405.26/) and to Mouse reference genome GRCm38 GCA_000001635.2 (https://www.ncbi.nlm.nih.gov/assembly/GCF_000001635.20/). Source data are provided with this paper.

## Code availability

All software that was used is free and open source and details on acquiring them can be found in the associated references. Code used to process data and generate figures in this study has been made available on GitLab (https://gitlab.com/cruk-mi/max-schenk-gucy/).

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

## Acknowledgements

Sample collection was undertaken via the ChemoRes Trial (molecular mechanisms underlying chemotherapy resistance, therapeutic escape, efficacy and toxicity—improving knowledge of treatment resistance in patients with lung cancer). Research samples were obtained from the Manchester Cancer Research Centre (MCRC) Biobank, UK. The role of the MCRC Biobank is to distribute samples and, therefore, cannot endorse studies performed or the interpretation of result. The work was funded by Cancer Research UK (CRUK) via core-funding to the CRUK Manchester Institute (grant no. A27412) and the CRUK Manchester Centre (grant no. A25254), and supported by the CRUK Manchester Experimental Cancer Medicines Centre (grant no. A20465), the CRUK Lung Cancer Centre of Excellence (grant no. A25146), and the NIHR Manchester Biomedical Research Centre. Figures 1a, 3a, and 5a were created with BioRender.com. The authors thank the Molecular Biology Core Facility, Histology, Scientific Computing, and BRU core facility for scientific support as well as Iain Hagan, Angeliki Malliri and Sir Salvador Moncada for constructive comments on the manuscript and Ekram Aidaros-Talbot for administrative support.

## Author contributions

Conception and design: M.W.S., K.K.F., and C.D. Development of methodology: M.W.S., M. Galvin, K.K.F., and C.D. Acquisition of data: M.W.S., M.R., S.P., A.L., S. Brown, and S. Bratt. Analysis and interpretation of data: M.W.S., S.H., A.S.M.M.H., T.D., C.Z., S.P.P., and A.K. Writing, review and/or revision of the manuscript: M.W.S., K.K.F., and C.D. Administrative, technical, or material support: F.B., L.P., M. Greenhalgh, and A.C. Study supervision: K.K.F., F.B., and C.D.

## Competing interests

The authors declare no competing interests.

## Additional information

**Peer review information** *Nature Communications* thanks Se Jin Jang, Takashi Kohno and the other anonymous reviewer(s) for their contribution to the peer review this work. Peer reviewer reports are available.

