## [Peer Review File · Nature Communications]

Reviewers' Comments:

Reviewer #1:

Remarks to the Author:

Comments:

In this manuscript by Schenk et al, they describe a new acquired resistance model of small cell lung cancer after etoposide chemotherapy using 6 pairs of longitudinal CDX model from small cell lung cancer patients. The author specified GUCY1B1 and GUCY1A1 as recurrent differentially expressed genes between pre-treatment models and progression models, and identified molecular mechanism of soluble guanylate cyclase signalling mediated acquired chemoresistance. I believe the most valuable aspect of this work is the identification of potential molecular targets of relapsed small cell lung cancer. The study should be eventually published after address of some important concerns.

Comment 1. The author did not show genomic alteration data of CDX17 and CDX7p, which are most widely used models for molecular mechanism studies. The genomic data should be presented.

Comment 2. The author described that they identified typical SCLC-associated mutation in their CDX models. However, mutational profiles of CDX18 and CDX20 presented in Fig.1 and Fig. S1 are quite different from typical SCLC-associated mutation: absence of TP53 and RB1 mutations, low total mutation numbers and low smoking related signature (C>A mutation). The author should exclude the possibility of other type of lung cancer resembling small cell carcinoma.

Comment 3. Photos of immunostaining in Fig 2 and Fig 5 should be changed with high magnification image to see cellular detail.

Reviewer #2:

Remarks to the Author:

The manuscript by Schenk et al. proposes an intriguing mechanism of chemoresistance in small cell lung cancer (SCLC). The Authors study unique clinical material collected before and after therapy and perform genomic and transcriptomic analysis. The results presented suggest that GUCY1B1 GUCY1A1 is upregulated in resistance tumors. In the further analysis, the authors provide evidence of transcriptional GUCY1B1 upregulation by Notch signaling and activation by nitric oxide (NO) signaling. Expression of GUCY1B1 in non-NE cells is consistent with Notch activity in those cells (Nature, 2017) and non-NE cells resistance to chemotherapy. However, several concerns need to be addressed before consideration for publication:

-Please provide high magnification pictures of IHCs for GUCY1B1. Are all cells equally positive/negative for GUCY1B1. The authors should provide co-IF staining for GUCY1B1 and Notch/Hes1. The non-NE and NE compartments within SCLC should be clear (see Lim et al. Nature, 2017)

-DAPT gamma-secretase is used to inhibit Notch receptor cleavage upon activation - the authors should use a direct genetic approach to downregulate Notch receptors to provide supporting evidence.

-HES1 expression is used as a readout of Notch activity but only HES2 was found to be upregulated in SCLC in response to therapy. Can authors provide evidence that HES2 is downregulated by DAPT or genetic modulation of Notch receptors? The regulation of HES2 by Notch is not as clear as for HES1. Is there any other evidence that Notch signaling is upregulated in chemo-resistant CDX?

-The Authors should provide direct evidence that active notch activates expression of GUCY1B1 / GUCY1A1 for e.g. though ChIP analysis.

-The Authors should provide additional control experiments for cisplatin, etoposide and doxorubicin resistance: H196 cells depleted for GUCY1B1 (as in Fig4 C) and stimulated with DETA NONOate (as in Fig 4D/F).

- in SCLC or in Xenograft what is the source of NO – is it endogenously produced by cancer cells is there evidence that the production is elevated?

-To validate that GUCY1B1 is acquired mechanisms of resistance the authors should seek to recapitulate in controlled conditions in vitro or in vivo if naïve cells e.g. CDX17 cells upregulated GUCY1B1 once the cells become resistant. That would also help to understand the dynamics of this process.

Reviewer #3:

Remarks to the Author:

The authors examined their original circulating tumor cell – derived explant (CDX) pairs obtained at pre and post chemotherapeutic timing and investigated a mechanism of “acquired” chemo-resistance of SCLC. They used their original (famous) resource and their approach is in contrast to that of previous studies, in which “inherent” chemo-resistance was investigated. They found that up-regulation of soluble guanylate cyclase (sGC) signaling is responsible for chemo-resistance of SCLC. The results are novel and have therapeutic implication for the miserable disease. The followings are the reviewer’s comments.

1. Figure 1a: Chemo-resistance of CDXs (P-series) is described only sketchy. Can the authors present real in vivo experimental data?

2. Genetic deleterious alterations of NOTCH1-3 genes have been observed in SCLC (George et al, Nature, 2015). Did the authors detected these alterations in CDXs and cell lines used in the present study? They might be involved in the acquisition of chemo-resistance.

3. NE and non-NE characters of CDXs ex vivo fractions (such as in Figure 3f-h); and H196 and H1048 cells should be validated by examining expression of neural markers.

4. PKG and NOS inhibitors might be applicable as a therapeutic agent for SCLC with acquired resistance by combining with other cytotoxic drugs. The reviewer recommends the authors examine the efficacy of these inhibitors for growth suppression of CDX (P-series) in vivo.

RESPONSE TO REVIEWERS

Reviewer #1 (Remarks to the Author):

Comments:

In this manuscript by Schenk et al, they describe a new acquired resistance model of small cell lung cancer after etoposide chemotherapy using 6 pairs of longitudinal CDX model from small cell lung cancer patients. The author specified GUCY1B1 and GUCY1A1 as recurrent differentially expressed genes between pre-treatment models and progression models, and identified molecular mechanism of soluble guanylate cyclase signalling mediated acquired chemoresistance. I believe the most valuable aspect of this work is the identification of potential molecular targets of relapsed small cell lung cancer. The study should be eventually published after address of some important concerns.

Comment 1: The author did not show genomic alteration data of CDX17 and CDX7p, which are most widely used models for molecular mechanism studies. The genomic data should be presented.

Response: We appreciate the comment about the missing whole exome sequencing (WES) data for CDX17 and CDX17P. We have now acquired the WES data for these models and included the data in Figure 1b-d, Supplementary Figure 1a-c as well as Supplementary Figure 2g in the revised manuscript. Although the addition of CDX17/CDX17P did not change the overall interpretation of the WES data, we did modify the manuscript as follows:

- In addition to identifying dysferlin (*DYSF*) as reported in our initial submission, analysis of recurrently acquired mutations including CDX17/CDX17P detected two additional mutations that were shared between multiple progression models: F-Box protein 10 (*FBXO10*) and cilia and flagella associated protein 47 (*CFAP47*). *In silico* analysis predicts that all three of these are passenger mutations (Tamborero *et al.*, Genome Med, 2018). We included these data in the revised version of the manuscript (page 5, lines 113 – 118).
- Including CDX17/CDX17P resulted in a trend ($p = 0.06$) of increased private mutations in progression models implicating acquisition of mutations through either SCLC progression or a mutagenic effect associated with exposure to chemotherapy. We amended the revised manuscript accordingly (page 6, lines 144 – 149).

Comment 2: The author described that they identified typical SCLC-associated mutations in their CDX models. However, mutational profiles of CDX18 and CDX20 presented in Fig.1 and Fig. S1 are quite different from typical SCLC-associated mutation: absence of TP53 and RB1 mutations, low total mutation numbers and low smoking related signature (C>A mutation). The author should exclude the possibility of other type of lung cancer resembling small cell carcinoma.

Response: The reviewer raised an important point that our initial WES analysis of CDX models derived from patient 18 and patient 20 did not display *TP53* and *RB1* mutations, the most commonly mutated

genes in SCLC. We therefore performed an experiment to investigate whether the p53 protein is functional and whether the RB1 protein is expressed in the aforementioned models:

P53 protein functionality and RB1 expression across CDX models.

- Ex vivo* cultures for CDX18P and CDX20P as well as the p53 WT cell line A549 were treated with nutlin3A in order to activate p53 expression. Expression of p53 as well as downstream p21 expression were assessed by western blot, vinculin served as loading control.
- Western blot for RB1 expression across CDX models, GAPDH served as loading control.

The p53 WT cell line A549 showed the expected low steady-state levels of p53 followed by induction of both p53 and its downstream effector p21 upon treatment with nutlin3A. Conversely, *ex vivo* cultures of both CDX18P and CDX20P exhibited elevated steady-state levels of p53, a pattern typical of mutant p53 (Frum *et al.*, Subcell Biochem., 2014). Furthermore, treatment with nutlin3A did not induce either p53 or p21, indicating that the p53 protein is not functional in these CDX models. Western blot analysis revealed that neither CDX18/CDX18P nor CDX20/CDX20P express detectable RB1 protein (western blot data included for reviewers only). Further analysis of our WES data revealed germline missense *TP53* mutations in patient 18 and patient 20 and germline nonsense *RB1* mutations in patient 20. Since our analysis pipeline only reports somatic mutations, we initially did not detect these mutations for the corresponding CDX models. We have now amended the text and Figure 1b accordingly and addressed this data in the revised version of the manuscript (page 5, lines 105 – 109 and page 30, lines 800 - 803). We hypothesize that *RB1* expression could be epigenetically silenced in CDX18/CDX18P, since we did not detect RB1 expression in these models by western blot and cannot identify any mutations. With this new analyses included, we now report *TP53* mutations and *RB1* mutations in 100% and 83% of our paired CDX models, respectively.

Whereas CDX18/18P showed a high smoking related signature, CDX20/20P showed a low smoking related signature compared to other paired CDX models. Clinical records revealed that this patient received a sibling allogeneic stem cell transplant for the treatment of acute myeloid leukaemia (AML), which could explain the low smoking related signature. We addressed these data in the revised version of the manuscript (pages 6-7, lines 153 – 157).

Comment 3: Photos of immunostaining in Fig 2 and Fig 5 should be changed with high magnification image to see cellular detail.

Response: In the revised manuscript, we replaced the original images for immunostaining in Figure 2 and Figure 5 now displaying higher magnification images in order to show cellular detail as requested.

Reviewer #2 (Remarks to the Author):

The manuscript by Schenk et al. proposes an intriguing mechanism of chemoresistance in small cell lung cancer (SCLC). The Authors study unique clinical material collected before and after therapy and perform genomic and transcriptomic analysis. The results presented suggest that GUCY1B1 GUCY1A1 is upregulated in resistance tumors. In the further analysis, the authors provide evidence of transcriptional GUCY1B1 upregulation by Notch signaling and activation by nitric oxide (NO) signaling. Expression of GUCY1B1 in non-NE cells is consistent with Notch activity in those cells (Nature, 2017) and non-NE cells resistance to chemotherapy. However, several concerns need to be addressed before consideration for publication:

Comment 1: Please provide high magnification pictures of IHCs for GUCY1B1. Are all cells equally positive/negative for GUCY1B1. The authors should provide co-IF staining for GUCY1B1 and Notch/Hes1. The non-NE and NE compartments within SCLC should be clear (see Lim et al. Nature, 2017)

Response: We appreciate the reviewer's comment to provide high magnification images. In the revised manuscript, we changed images for immunostainings in Figure 2 and Figure 5 to display higher magnification images. Furthermore, we performed co-IF staining for GUCY1B1 and HES1 in NE and Non-NE cell fractions that were separated in CDX17 and CDX17P and we confirmed the NE and Non-NE status of different cell populations by staining for synaptophysin (SYP) and REST:

GUCY1B1 and HES1 co-immunofluorescence in NE and Non-NE CDX17 and CDX17P.

- a) IHC of CDX17 and CDX17P NE and Non-NE CDX cells for synaptophysin (SYP) and REST. Scale bar set to 50 μ m and equivalent throughout panels.
- b) Co-IF for GUCY1B1 and HES1 in CDX17 and CDX17P NE and Non-NE CDX cells. Scale bar set to 50 μ m and equivalent throughout panels. Insets show higher magnification images.

Performing co-IF on NE and Non-NE cells of CDX17 and CDX17P, we were able to observe an expected increase of HES1 in Non-NE cells, concomitant with an increase in GUCY1B1 expression. Furthermore, we could detect cells co-expressing HES1 and GUCY1B1, providing further evidence that cells with active Notch signalling have relatively high expression of GUCY1B1. We included these data in Supplementary Figure 3d-e and we discuss the data in the revised version of the manuscript (page 13, lines 363 – 366).

Comment 2: DAPT gamma-secretase is used to inhibit Notch receptor cleavage upon activation - the authors should use a direct genetic approach to downregulate Notch receptors to provide supporting evidence.

Response: The reviewer raised an important point that the gamma-secretase inhibitor DAPT inhibits Notch activation by interfering with Notch receptor cleavage. As suggested by the reviewer, we used CRISPR to genetically downregulate Notch1 and investigated GUCY1B1 expression by western blot:

Genetic downregulation of Notch1 expression using CRISPR. Western blot for Notch1, HES1, and GUCY1B1 in sgNTA, sgNotch1-1, and sgNotch1-2 H1048 cells (n=3).

Downregulation of Notch1 using two different sgRNAs resulted in reduction of Notch signalling (HES1 expression) as well as the expression of GUCY1B1. Whereas sgNotch1-1 cells showed a high reduction of Notch1 expression, sgNotch1-2 cells showed a lower reduction of Notch1 expression. Reduced GUCY1B1 expression was consistent with the degree of reduction of Notch1 expression. We included these new data in Figure 3e and in the revised version of the manuscript (pages 11 – 12, lines 293 – 297).

Furthermore, in order to control for off-target effects of DAPT, we repeated our experiments with the gamma-secretase inhibitor DBZ in CDX *ex vivo* cultures as well as SCLC cell lines:

Treatment with the gamma-secretase inhibitor DBZ leads to downregulation of Notch signalling and GUCY1B1/GUCY1A1 expression.

- HEY1, GUCY1B1 and GUCY1A1 mRNA expression measured by RT-qPCR of DMSO- or DBZ-treated H196 cells. Fold enrichment normalized to untreated control and Beta-2-Microglobulin (B2M) housekeeping gene expression. n = 3, data are represented as mean ± SEM. p values from two-sided unpaired Student's t test following Shapiro-Wilk test to confirm normality. HEY1 (t = 16.14, df = 4), GUCY1B1 (t = 8.608, df = 4), GUCY1A1 (t = 16.06, df = 4).
- Western blot for GUCY1B1 and HES1 expression after treatment of H196 cells with DMSO or DBZ. Quantification of GUCY1B1 (relative volume intensity, RVI) on the right, normalized to H3 expression. n = 3, data are represented as mean ± SEM; p values from two-sided unpaired Student's t test following Shapiro-Wilk test to confirm normality. GUCY1B1 (t = 9.368, df = 4), HES1 (t = 7.553, df = 4).
- HEY1, GUCY1B1 and GUCY1A1 mRNA expression measured by RT-qPCR of DMSO- or DBZ-treated CDX17P Non-NE cells. Fold enrichment normalized to untreated control and Beta-2-Microglobulin (B2M) housekeeping gene expression. n = 3, data are represented as mean ± SEM. p values from two-sided unpaired Student's t test following Shapiro-Wilk test to confirm normality. HEY1 (t = 7.1711, df = 4), GUCY1B1 (t = 3.744, df = 4), GUCY1A1 (t = 4.111, df = 4).

Using the gamma-secretase inhibitor DBZ, we recapitulated our results with DAPT, strengthening the data set that shows that inhibition of Notch signalling leads to downregulation of sGC subunit expression. Consequently, these results provide further evidence that Notch signalling regulates the

expression of sGC subunits, and we included these data in Supplementary Figure 3a-c and discussed the data in the revised manuscript (page 11, lines 286 – 293 and page 13, lines 358 – 361).

Comment 3: HES1 expression is used as a readout of Notch activity but only HES2 was found to be upregulated in SCLC in response to therapy. Can authors provide evidence that HES2 is downregulated by DAPT or genetic modulation of Notch receptors? The regulation of HES2 by Notch is not as clear as for HES1. Is there any other evidence that Notch signaling is upregulated in chemo-resistant CDX?

Response: We appreciate the reviewer's comment that the regulation of HES2 by Notch signalling is not as clear as for HES1. We therefore treated SCLC cell lines with DAPT and investigated the expression of HES2 by qPCR as suggested by the reviewer. Furthermore, we overexpressed N1ICD in SCLC cell lines and studied HES2 expression by qPCR:

HES2 expression is not changed by DAPT treatment or overexpression of N1ICD. HEY1 and HES2 mRNA expression measured by RT-qPCR of a) DMSO- or DAPT-treated cells or b) after overexpression of N1ICD. Fold enrichment normalized to untreated control and Beta-2-Microglobulin (*B2M*) housekeeping gene expression. n = 3, data are represented as mean ± SEM. p values from two-sided unpaired Student's t test following Shapiro-Wilk test to confirm normality.

Treatment with the gamma-secretase inhibitor DAPT led to a downregulation of HEY1 as observed previously, however HES2 expression did not change. Furthermore, overexpression of N1ICD led to an upregulation of HEY1 expression as expected, whereas HES2 expression remained unchanged (data included for reviewers only). Therefore, in our hands HES2 cannot serve as a marker of active Notch signalling and we provide alternative evidence that Notch signalling is upregulated in our progression CDX models compared to baseline models. In our previous publication by Simpson *et al.* (Nature Cancer, 2020), we show the upregulation of Notch receptors in four out of six disease progression CDX pairs. Furthermore, differential gene expression analysis revealed the downregulation of two negative regulators of Notch signalling in our progression models: BEN domain containing 6 (*BEND6*) and delta-like 1 homolog (*DLK1*) were downregulated 2-fold and 3-fold (p<0.001), respectively, across our progression models. Both *BEND6* (Dai *et al.*, 2013, Development) and *DLK1* (Finn *et al.*, 2019, Cell

Reports) were shown to be negative regulators of Notch signalling and their downregulation indicates upregulation of Notch signalling in our progression models. We amended the text in this paragraph in the revised version of the manuscript accordingly (page 11, lines 278 – 284).

Comment 4: The Authors should provide direct evidence that active notch activates expression of GUCY1B1 / GUCY1A1 for e.g. though ChIP analysis.

Response: The reviewer made an important suggestion to provide evidence that Notch directly activates the expression of the sGC subunits. We therefore performed chromatin immunoprecipitation (ChIP)-qPCR in a cell line overexpressing the Notch1 intracellular domain (N1ICD) by performing ChIP using a Notch1 antibody followed by qPCR with primers flanking previously reported RBPJ-binding sites in the GUCY1B1 and GUCY1A1 promoter. As a positive control, we investigated N1ICD binding to the HES1 promoter:

Notch1 ChIP-qPCR in H1048 cells overexpressing N1ICD. qPCR of RBPJ binding sites in the GUCY1A1 and GUCY1B1 promoter was performed using primers flanking RBPJ binding sites. N1ICD binding to the HES1 promoter served as positive control, whereas binding to a negative control region served as negative control. n = 3, data are represented as mean \pm SD. p values from two-sided unpaired Student's t test following Shapiro-Wilk test to confirm normality.

ChIP followed by qPCR of previously reported RBPJ binding sites in the GUCY1A1 and GUCY1B1 promoters demonstrated that N1ICD bound both promoter regions as well as the expected binding in the HES1 promoter. Our results therefore indicate that GUCY1A1 and GUCY1B1 are Notch target genes in SCLC. We included these new data in Figure 3j in the revised manuscript and added new text describing this finding (pages 13 - 14, lines 366 – 373).

Comment 5: The Authors should provide additional control experiments for cisplatin, etoposide and doxorubicin resistance: H196 cells depleted for GUCY1B1 (as in Fig4 C) and stimulated with DETA NONOate (as in Fig 4D/F).

Response: Using both genetic and pharmacological approaches, we were able to demonstrate that downregulation of sGC signalling can sensitize cells to chemotherapy with etoposide and doxorubicin. The suggested experiment is already included in Figure 4c: H196 cells were depleted for GUCY1B1 and stimulated with DETA NONOate, and knockdown of GUCY1B1 resulted in increased chemosensitivity (Figure 4c). As expected, GUCY1B1 depletion alone without activation of sGC signalling through DETA NONOate did not result in a change in chemosensitivity (Supplementary Figures 4e and 4f). We amended the manuscript to emphasize that cells in Figure 4c were stimulated with DETA NONOate prior to treatment with chemotherapy (page 15, lines 398 – 400).

Comment 6: In SCLC or in Xenograft what is the source of NO – is it endogenously produced by cancer cells is there evidence that the production is elevated?

Response:

The reviewer raised an intriguing question on the source of NO in our CDX xenograft models. We feel this question is beyond the scope of our current study but future work with NO signalling experts outside the cancer field will focus on identifying the cellular source of NO.

Comment 7: To validate that GUCY1B1 is acquired mechanisms of resistance the authors should seek to recapitulate in controlled conditions in vitro or in vivo if naïve cells e.g. CDX17 cells upregulated GUCY1B1 once the cells become resistant. That would also help to understand the dynamics of this process.

Response: The reviewer made an important suggestion to investigate GUCY1B1 expression in CDX cells in which acquired chemoresistance was generated *in vitro* or *in vivo*. Using an approach similar to that published by Gardner *et al.* (Cancer Cell, 2017), we challenged CDX17 with chemotherapy *in vivo* until it no longer responded to chemotherapy. Following this strategy, we were able to generate paired chemo-naïve and resistant CDX models. We subsequently compared GUCY1B1 expression in these paired models performing western blots:

Generation of acquired chemoresistance by re-challenge of chemo-naive CDX models *in vivo*. CDX17 was challenged with chemotherapy for up to three cycles (dashed lines indicate treatment cycle). Cumulative toxicity necessitated the model to be passaged to a new generation of mice after three cycles. Red lines indicate mice treated with cisplatin/etoposide, and blue lines vehicle treated mice. Panel on the right indicates western blot on tumour lysates for GUCY1B1. For western blot, two replicate tumour pieces were analyzed.

Following a strategy of chemotherapy serial challenge of mice bearing CDX17, we were able to generate resistant derivatives (data included for reviewers only). We did not observe an upregulation of GUCY1B1 expression by serial challenge of CDX17, despite recurrent GUCY1B1 upregulation in our patient derived models. Similarly, we did not identify recurrently decreased *SLFN11* expression in our CDX progression models in contrast to a previous study in SCLC (Gardner *et al.*, Cancer Cell, 2017). The observed differences could be explained by the different model systems used. The advantage of patient-derived paired models of resistance is that it enables investigation of acquired resistance occurring in the individual patient, as opposed to acquired chemoresistance generated through chemotherapy re-challenge in the lab, an approach commonly taken, but that may not always reflect the biology of a patients progressing tumour, as shown here.

Reviewer #3 (Remarks to the Author):

The authors examined their original circulating tumor cell – derived explant (CDX) pairs obtained at pre and post chemotherapeutic timing and investigated a mechanism of “acquired” chemo-resistance of SCLC. They used their original (famous) resource and their approach is in contrast to that of previous studies, in which “inherent” chemo-resistance was investigated. They found that up-regulation of soluble guanylate cyclase (sGC) signaling is responsible for chemo-resistance of SCLC. The results are novel and have therapeutic implication for the miserable disease. The followings are the reviewer’s comments.

Comment 1: Figure 1a: Chemo-resistance of CDXs (P-series) is described only sketchy. Can the authors present real *in vivo* experimental data?

Response: We appreciate the reviewer’s comment and would like to refer to Supplementary Table 1 for *in vivo* chemosensitivity of our paired CDX models as well as to our previous publication (Simpson *et al.*, 2020) for detailed *in vivo* experimental data.

Comment 2: Genetic deleterious alterations of NOTCH1-3 genes have been observed in SCLC (George et al, Nature, 2015). Did the authors detected these alterations in CDXs and cell lines used in the present study? They might be involved in the acquisition of chemo-resistance.

Response: The reviewer raised an important point to investigate mutations in Notch 1-3 which have been previously observed in SCLC. We therefore analyzed our WES data set in regards to somatic mutations in Notch1-3:

Somatic Notch receptor mutations in CDX models according to WES data.

CDX Model	Gene	Chromosome	Variant Classification
CDX3	NOTCH3	19	Intron
CDX3P	NOTCH3	19	Intron
CDX20	NOTCH3	19	Intron
CDX20	NOTCH3	19	Silent
CDX20	NOTCH3	19	Intron
CDX20P	NOTCH1	9	Intron
CDX20P	NOTCH3	19	Silent
CDX20P	NOTCH3	19	Intron

Investigating somatic mutations in Notch1-3, we did not detect inactivating mutations for Notch1-3 and we only identified intronic mutations in Notch3 for CDX3, CDX3P, CDX20, and CDX20P as well as a silent mutation in Notch3 for CDX20 and CDX20P (data included for reviewers only). Furthermore, we investigated Notch1-3 mutations in H1048 and H196 cell lines using CellMiner (Tlemsani *et al.*, Cell Reports, 2020). Using the CCLE-Broad-MIT and GDSC-MGH-Sanger platforms as references, we did not detect mutations in Notch1-3.

Comment 3: NE and non-NE characters of CDXs *ex vivo* fractions (such as in Figure 3f-h); and H196 and H1048 cells should be validated by examining expression of neural markers.

Response: The reviewer made an important suggestion to validate the NE and Non-NE characters of CDX *ex vivo* fractions as well as cell lines used in the manuscript. We therefore performed western blot analysis for the neuroendocrine marker synaptophysin (SYP). We included these data in the revised manuscript in Figure 3b. NE and Non-NE characters of CDX *ex vivo* cultures used in the current study were validated in Figure 3g and discussed in the manuscript (page 13, lines 356 – 358) and confirm our previous publication (Pearsall *et al.*, Journal of Thoracic Oncology, 2020).

Comment 4: PKG and NOS inhibitors might be applicable as a therapeutic agent for SCLC with acquired resistance by combining with other cytotoxic drugs. The reviewer recommends the authors examine the efficacy of these inhibitors for growth suppression of CDX (P-series) *in vivo*.

Response: We appreciate the reviewer's suggestion to examine the efficacy of PKG and NOS inhibitors for growth suppression of our CDX progression models *in vivo*. Due to the unavailability of potent PKG inhibitors for *in vivo* use, we decided to test the efficacy of L-NMMA, a NOS inhibitor which has been shown to reduce tumour growth in triple-negative breast cancer mouse models (Granados-Principal *et al.*, 2015). We chose to test the NOS inhibitor in CDX17P, the same CDX progression model used for gene editing experiments (Figure 5i and j). Similar to the study by Granados-Principal *et al.* and in order to increase tolerability to the compound, we implanted SCID Beige mice with CDX17P. Mice were either

treated with standard of care cisplatin/etoposide, L-NMMA, or the combination cisplatin/etoposide + L-NMMA. Our dosing regimen was guided by the study of Granados-Principal *et al.*, treating mice with 400 mg/kg L-NMMA on day 1 of dosing followed by four consecutive days of treatment with 200 mg/kg L-NMMA, which was the maximum tolerated dose of L-NMMA:

L-NMMA NOS inhibitor treatment of CDX17P.

- Mice implanted with CDX17P were treated with cisplatin/etoposide, L-NMMA, cisplatin/etoposide + L-NMMA or vehicle treated. Data are from 3-5 animals per experimental arm and represented as mean \pm SEM.
- Kaplan-Meier survival curve comparing percent event free survival of mice until the tumour reaches 4 x initial tumour volume (ITV). p value from log-rank test, chi square = 3.875, df = 1.

Treatment with cisplatin/etoposide elicited only a minor reduction in tumour growth which was significantly enhanced by L-NMMA co-treatment, resulting in increased event free survival (median 13 and 17 days in mice treated with cisplatin/etoposide or cisplatin/etoposide + L-NMMA, respectively, $p=0.049$). These results indicate that treatment of CDX17P with L-NMMA increased efficacy of cisplatin/etoposide, complementing and consistent with the data from the genetic reduction of GUCY1B1 expression in CDX17P. We included the new data in Figure 5i and j and discussed the data in the revised manuscript (page 19, lines 544 – 550 and page 22, lines 612 – 622).

Reviewers' Comments:

Reviewer #1:

Remarks to the Author:

Genomic alteration data of CDX17 and CDX7p are properly presented.

Absence of TP53 and RB mutations in CDX18 and CDX20 was explained properly by using alternative molecular studies.

Therefore, all my concerns have been resolved.

Reviewer #2:

Remarks to the Author:

I commend the Authors for providing several additional important experimental evidence and clarifications in the revised manuscript. However, there are two remaining issues that limit the significance of the manuscript, specifically:

(1) The Authors propose that soluble guanylate cyclase (sGC), the only known receptor for nitric oxide (NO), is critical in driving small cell lung cancer (SCLC) chemoresistance. Moreover, the Authors demonstrated conclusively that sGC requires nitric oxide (NO) to manifest its function in SCLC chemoresistance. Consequently, activation of the proposed resistance mechanism is directly dependent on nitric oxide. Hence my question about the source of NO, which drives the signaling activation, is pertinent. The Authors "feel this question is beyond the scope of our current study". I agree in part, but that brings a major limitation to the significance and relevance of the study. In particular in the context of the utilized model. Specifically, nitric oxide and sGC are involved in blood vessels dilation. The presented CDX model depends on collecting circulating tumor cells found in the blood of patients. Therefore, it is possible that the sGC pathway is activated in those circulating cells due to NO presence in blood vesicles, but what is the source of NO in primary tumor nodules? Perhaps NO availability is the reason why the Authors were not able to recapitulate sGC upregulation in xenograft (see point 2).

(2) The authors could not validate the proposed mechanisms of acquired resistance to chemotherapy in a controlled experimental model. As suggested in my review, the Authors used chemo-naïve CDX (CDX17) and generated resistant tumors using serial chemotherapy challenge of mice bearing CDX17 xenografts. However, the Authors have not observed changes in GUCY1B1 expression once the resistance was acquired. Therefore the phenomenon of GUCY1B1 upregulation upon chemotherapy is currently limited to an observation made in circulating tumor cells in 4 out of 6 patients and cannot be experimentally modeled. The Authors argue that the inability to recapitulate this mechanism shows "the advantage of patient-derived paired models of resistance". However, in my opinion, that is to be determined here and is indeed a limitation to accept the proposed mechanisms of acquired SCLC chemoresistance.

Reviewer #3:

Remarks to the Author:

The manuscript has been properly revised according to the reviewer's comments.

Reviewer 2 comments

1. *“The Authors propose that soluble guanylate cyclase (sGC), the only known receptor for nitric oxide (NO), is critical in driving small cell lung cancer (SCLC) chemoresistance. Moreover, the Authors demonstrated conclusively that sGC requires nitric oxide (NO) to manifest its function in SCLC chemoresistance. Consequently, activation of the proposed resistance mechanism is directly dependent on nitric oxide. Hence my question about the source of NO, which drives the signaling activation, is pertinent. The Authors “feel this question is beyond the scope of our current study”. I agree in part, but that brings a major limitation to the significance and relevance of the study. In particular, in the context of the utilized model. Specifically, nitric oxide and sGC are involved in blood vessels dilation. The presented CDX model depends on collecting circulating tumor cells found in the blood of patients. Therefore, it is possible that the sGC pathway is activated in those circulating cells due to NO presence in blood vesicles, but what is the source of NO in primary tumor nodules? Perhaps NO availability is the reason why the Authors were not able to recapitulate sGC upregulation in xenograft.”*

Response

It was good to hear that the reviewer agrees that we conclusively demonstrate a requirement for NO in the role of sGC in SCLC acquired chemoresistance. We completely agree with him/her that the source of NO in our CDX models is a pertinent question, but we strongly disagree that the relevance of the manuscript is severely diminished without elucidation of the NO source.

As the reviewer points out and we agree, it is possible that NO in the blood activates sGC in CTCs. However, these CTCs have been engrafted in the mouse and the resultant tumour has been passaged through at least 3 animals to generate a CDX model. It seems highly unlikely then that the cells ‘remember’ the NO they encountered as CTCs in the donor patient as sGC molecules are inactivated due to oxidation and protein turnover (Stuehr *et al.*, Journal of Biological Chemistry 2021).

As CDX are highly vascularized, we propose that it is more likely that mouse endothelial cells release NO to activate sGC upregulated in SCLC CDX cells. There is at least one other intriguing possibility as we previously reported in Nature Communications that SCLC cells can undergo vasculogenic mimicry, making their own vessels, and adopting endothelial behaviours that could include NO production (Williamson *et al.*, Nature Comms 2016). This is, of course, highly speculative and warrants a future study, beyond the scope of this paper, as we are glad the editorial decision agrees.

On this point, we further highlight that the NOS inhibitor L-NMMA recapitulated the chemosensitizing phenotype induced by genetic targeting of sGC supporting the hypothesis that NO present in tumours can activate the sGC pathway to mediate chemoresistance.

We have added the following text to the discussion, attempting to speculate on the source of NO in attempt to alleviate the concern (page 20, lines 536 – 543 of the altered manuscript with no markup).

“An outstanding question is the source of NO within tumours. NO is typically synthesized by endothelial cells and SCLC is a well vascularised tumour³⁹ as are our CDX models, implicating the vasculature as a possible NO source. Alternatively, we have previously shown that SCLC cells can undergo vasculogenic mimicry, a process whereby cancer cells acquire properties of endothelial

cells and form vessel-like structures⁴⁰, raising the intriguing possibility that SCLC cells may generate NO themselves, a hypothesis warranting further study.”

2. The authors could not validate the proposed mechanisms of acquired resistance to chemotherapy in a controlled experimental model. As suggested in my review, the Authors used chemo-naïve CDX (CDX17) and generated resistant tumors using serial chemotherapy challenge of mice bearing CDX17 xenografts. However, the Authors have not observed changes in GUCY1B1 expression once the resistance was acquired. Therefore the phenomenon of GUCY1B1 upregulation upon chemotherapy is currently limited to an observation made in circulating tumor cells in 4 out of 6 patients and cannot be experimentally modeled. The Authors argue that the inability to recapitulate this mechanism shows “the advantage of patient-derived paired models of resistance”. However, in my opinion, that is to be determined here and is indeed a limitation to accept the proposed mechanisms of acquired SCLC chemoresistance.

Response

The reviewer correctly states that we were unable to recapitulate GUCY1B1 upregulation via a system in which a chemosensitive baseline model is serially re-challenged in the laboratory. Our preferred experimental system utilizes patient derived models in which we study the resistance that occurred in the patient because this surely more accurately reflects ‘ground truth’ than resistance generated in the laboratory setting where it is difficult to accurately recapitulate the clinical setting (e.g., dose modifications, patient-relevant drug exposure, etc.). Generation of PDX models from drug resistant patients has previously been used to identify clinically relevant mechanisms of acquired drug resistance (Krumbach *et al.*, European Journal of Cancer, 2011). Conversely, decades of attempts to discover mechanisms of chemoresistance in SCLC via repeated increased drug dosing of treatment naïve chemosensitive models failed to identify mechanisms that were then validated in patients (Scientific Framework for SCLC, NCI, 2014). More recently Gardner *et al.* (Cancer Cell, 2017) observed the recurrent downregulation of *SLFN11* expression in models with laboratory driven acquired chemoresistance and there is evidence that this may be clinically relevant, however neither Drapkin *et al.* (Cancer Discovery, 2018) using patient derived models nor Wagner *et al.* (Nature Communications, 2018) using patient tissue samples could validate *SLFN11* as a mechanism of chemoresistance. Overall, these results suggest that there are multiple mechanisms of SCLC acquired chemoresistance and that different models and experimental approaches may reveal diverse molecular mechanisms. Our stance is that both approaches are valuable tools to discover novel mechanisms of acquired chemoresistance, and that all discoveries will require further clinical validation.

We have modified and extended the paragraph on different approaches to discover acquired resistance mechanisms to more fully discuss the disparities that occur and strengths and weaknesses of each approach to give a more balanced view (page 22- 23, lines 587 – 615 of the altered manuscript with no markup) as follows:

“There are multiple, valid approaches to uncover mechanisms of acquired resistance in SCLC. A commonly used approach is to repeatedly treat mouse models (cell line xenografts or PDX) with increasing drug concentrations until resistance emerges. Transcriptional downregulation of

SLFN11 has been identified as one mechanism of SCLC acquired chemoresistance using this drug re-challenge strategy to study ten PDX mouse models driven to chemotherapy resistance⁵. However, comparing *SLFN11* transcript and protein expression levels in CDX and PDX models, Drapkin *et al.* did not identify a difference between models derived from untreated and previously treated patients⁷ and we could also not observe decreased *SLFN11* in our progression CDX. Another approach is to interrogate patient biospecimens to identify alterations that correlate with acquired resistance, although this approach is limited by scarce availability of paired pre- and post-treatment biopsies and an inability to function test candidate mechanisms in the clinical material. Wnt-pathway genes⁴ have been implicated in SCLC resistance via WES performed on paired tumour samples from 12 SCLC patients; we did not observe these changes in our CDX study.

Our study for the first time employs transcriptomic analysis on six paired SCLC patient-derived baseline and progression models to uncover mechanisms of acquired chemoresistance, where in contrast to previous mouse modelling studies, acquired resistance occurred in the patient.

These disparities in studies on acquired resistance mechanisms could be explained by the different model systems. Furthermore, we did not detect *GUCY1B1* and *GUCY1A1* upregulation in all six CDX progression models. We did not expect sGC signalling to be the sole reason for chemoresistance given the high degree of ITH in relapsed SCLC (demonstrated by single cell RNA-seq analysis in eight CDX models¹¹) that infers likely existence of multiple chemoresistance mechanisms. We conclude that whichever approach is used to discover mechanisms of acquired chemoresistance, and we favour the paired CDX approach, *in vivo* validation followed by clinical testing are the required next steps.”